# RENGE infers gene regulatory networks using time-series single-cell RNA-seq data with CRISPR perturbations

Masato Ishikawa [1✉], Seiichi Sugino[1], Yoshie Masuda[1], Yusuke Tarumoto [1], Yusuke Seto[1], Nobuko Taniyama[1], Fumi Wagai[1], Yuhei Yamauchi[1], Yasuhiro Kojima [2], Hisanori Kiryu[3], Kosuke Yusa [1], Mototsugu Eiraku [1,4] & Atsushi Mochizuki[1]

Single-cell RNA-seq analysis coupled with CRISPR-based perturbation has enabled the inference of gene regulatory networks with causal relationships. However, a snapshot of single-cell CRISPR data may not lead to an accurate inference, since a gene knockout can influence multi-layered downstream over time. Here, we developed RENGE, a computational method that infers gene regulatory networks using a time-series single-cell CRISPR dataset. RENGE models the propagation process of the effects elicited by a gene knockout on its regulatory network. It can distinguish between direct and indirect regulations, which allows for the inference of regulations by genes that are not knocked out. RENGE therefore outperforms current methods in the accuracy of inferring gene regulatory networks. When used on a dataset we derived from human-induced pluripotent stem cells, RENGE yielded a network consistent with multiple databases and literature. Accurate inference of gene regulatory networks by RENGE would enable the identification of key factors for various biological systems.

[1] Institute for Life and Medical Sciences, Kyoto University, Kyoto 606-8507, Japan. [2] Laboratory of Computational Life Science, National Cancer Center Research Institute, Tokyo 104-0045, Japan. [3] Department of Computational Biology and Medical Sciences, Graduate School of Frontier Sciences, The University of Tokyo, Kashiwa, Chiba 277-8561, Japan. [4] Institute for the Advanced Study of Human Biology (WPI-ASHBi), Kyoto University, Kyoto 606-8507, Japan. ✉email: ishikawa.masato.7v@kyoto-u.ac.jp

Understanding gene regulation is crucial for our comprehension of biological processes. As information on gene regulation has accumulated, it has become clear that the regulatory relationships between genes constitute complex systems, designated gene regulatory networks (GRNs)[1,2]. Hence, elucidating the structures of GRNs is expected to provide insights into system dynamics including the identification of key genes that control the behavior of the entire system[3–5].

GRNs have been inferred using a variety of methods[6,7], which are categorised into three groups based on data used. The first group utilizes the binding of transcription factors (TFs) to DNA. This method assesses the regulatory relationships between genes via the binding of TFs to regulatory regions on the genome, which can be inferred using the TF binding motifs, or detected directly with ChIP-seq[8]. However, binding does not necessarily indicate regulation, and assessing the binding of numerous genes via ChIP-seq is a laborious undertaking. The second group uses co-expression relationships of genes under unperturbed conditions. The generation of large-scale gene expression data has spurred the development of numerous methods to infer GRNs by examining the relationships between individual gene expression. For example, GENIE3[9] is a leading method in this field and boasts outstanding performance in the DREAM benchmark[10] and in the analysis of GRN inference from scRNA-seq data[11]. Although co-expressed genes can be estimated based on observational expression data with no genetic perturbation, it is generally difficult to infer causality, or regulation, between genes based solely on these data.

The third group relates to changes in gene expression upon genetic perturbation. The most reliable way to infer a causal relationship between genes would be to examine the changes in expression resulting from perturbed gene expression[12]. In particular, the recently developed single-cell CRISPR (scCRISPR) analysis[13–15] and associated computational tools, including MIMOSCA[13] and scMAGeCK[16], have made it possible to infer regulatory relationships between genes by measuring changes in expression resulting from the knockout (KO), or knockdown, of a relatively large number of genes.

These analysis methods use expression data collected from a given snapshot following gene KO to infer the regulatory effects elicited by the KO gene on other genes. Hence, via detection of expression changes following KO, these methods can effectively detect causal relationships; however, these methods are limited by their measurement of only gene expression snapshots (Fig. 1a). In particular, these methods are limited in their ability to distinguish between direct and indirect regulation. That is, when a gene is knocked out, changes in expression may occur in the genes directly regulated by the KO gene as well as those further downstream. Thus, existing methods cannot effectively determine whether the expression changes are due to direct regulation from the KO gene or indirect regulation via other genes. Additionally, these methods only infer regulation caused by KO genes, without accounting for the potential regulation elicited by genes that were not knocked out (non-KO genes). As such, all genes in the focal system must be knocked out to obtain a complete GRN.

Here, we address these shortcomings by integrating time-series scCRISPR analysis and a newly developed computational method for GRN inference. Given that the sequential changes in expression occurring after gene KO should reflect gene regulatory relationships, we propose the measurement of gene expression at multiple time points to distinguish early and late changes occurring after gene KO with the CRISPR system (Fig. 1b). To take full advantage of such dataset, we developed RENGE (REgulatory Network inference using GEne perturbation data), a computational method to infer GRNs from time-series expression data after gene KO (Fig. 1c). More specifically, RENGE models the process through which the KO effects are propagated on the network. Moreover, it can distinguish direct and indirect regulation more accurately than existing methods and infer regulation by non-KO genes.

Using data generated by the GRN-based simulator dyngen and scCRISPR data of human induced pluripotent stem cells (hiPSCs), we show that RENGE outperforms the existing methods in its ability to infer GRNs. We then use the GRN of hiPSCs inferred by RENGE to predict gene pairs that function as a protein complex, which are further validated using multiple databases. These analyses suggest a previously unknown key factor for pluripotency maintenance, namely, a PRDM14 and RUNX1T1 complex. Finally, we demonstrate that RENGE can utilize the inferred GRN to predict changes in the expression of other genes after the KO of any gene in the network.

## Results

**Algorithm overview.** We have developed the RENGE method to infer GRNs from time-series scCRISPR analysis data (Fig. 1c). Suppose we have expression data $\mathbf{E}_{g,t}$ of each gene in a cell observed at time $t$ after knocking out a gene $g$ in the cell. RENGE regresses the expression $\mathbf{E}_{g,t}$ of each gene at each time $t$ following the decrease $\mathbf{X}_g$ in KO gene $g$ expression. Consider the effects of a gene KO in a cell as it spreads stepwise in the GRN. Here, if the effect of a change in the expression of gene $j$ is propagated to gene $i$ via a path containing $k-1$ intermediate genes, we consider the effect to be due to $k$-th order regulation. The gene expression $\mathbf{E}'_{g,K'}$, where the effect up to $K'$-th order regulation from the KO gene $g$ appears, can be modeled as follows:

$$\mathbf{E}'_{g,K'} = \sum_{k'=1}^{K'} \mathbf{A}^{k'} \mathbf{X}_g + \mathbf{b}_{K'}, \qquad (1)$$

where $\mathbf{E}'_{g,K'}$ is the $G$-dimensional expression vector, $\mathbf{A}$ is the $G \times G$ matrix representing the GRN, and each element $\{\mathbf{A}\}_{i,j}$ $(i \neq j)$ represents the strength of regulation from gene $j$ to gene $i$, where $G$ is the number of genes. Only when $i = j$, does $\{\mathbf{A}\}_{i,j}$ represent multiple effects including degradation and self-regulation. For more details, see Supplementary Note 1. $\mathbf{X}_g$ is the $G$-dimensional vector representing the decrease in KO gene $g$ expression, and $\mathbf{b}_{K'}$ is the $G$-dimensional expression vector corresponding to the wild type. $\sum_{k'=1}^{K'} \mathbf{A}^{k'} \mathbf{X}_g$ represents the expression change from the wild type due to gene KO. Since the KO gene $g$ is no longer regulated by other genes, the row for the KO gene $g$ in $\mathbf{A}$ is set to 0 (see Methods for more detail). If we are given the gene expression $\mathbf{E}'_{g,K'} (K' = 1, \ldots, max\_K')$, in which the effects of different maximum order $K'$ of regulation appear, we can infer the GRN $\mathbf{A}$ by fitting the model (1) to the data.

However, the longest path length of regulatory interactions that have occurred, $K'$, is usually unknown at the measurement time $t$ of gene expression. To address this, RENGE introduces the term $w(t, k, g)$ that expresses the strength of the effect of the $k$-th order regulation from the KO gene $g$ at measurement time $t$. Estimating $w(t, k, g)$ from the data enables the estimation of $\mathbf{A}$ even when the maximum order of regulation is unknown at each measurement time $t$. Finally, the RENGE model equation is as follows:

$$\mathbf{E}_{g,t} = \sum_{k=1}^{K} w(t, k, g) \mathbf{A}^k \mathbf{X}_g + \mathbf{b}_t. \qquad (2)$$

Given the time-series expression data with the different KO genes $g$, RENGE estimates $\mathbf{A}$ so that the model fits the entire expression data set. Note that RENGE cannot infer self-regulation as the diagonal elements of $\mathbf{A}$ do not necessarily represent only self-regulation. The $p$-value for each element of the parameter $\mathbf{A}$

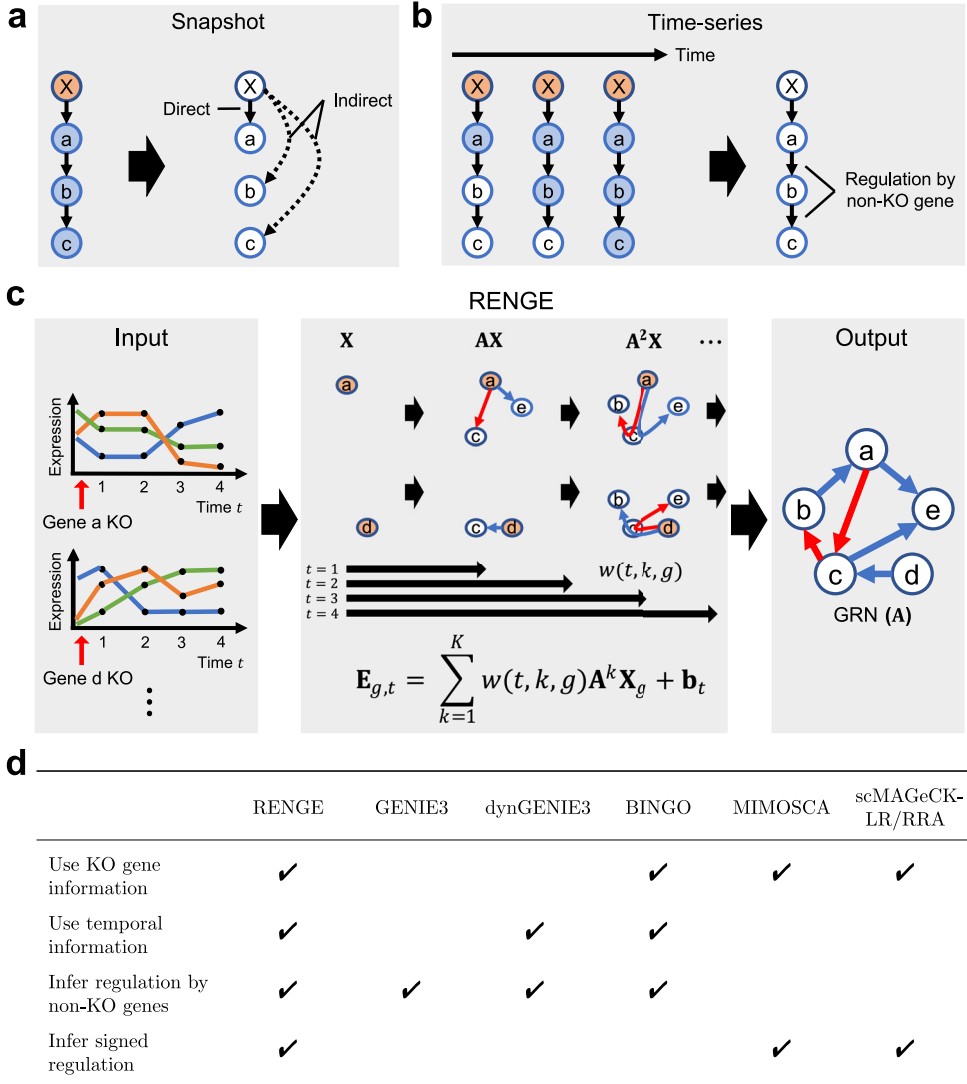

**Fig. 1 Overview of RENGE. a** The use of snapshot expression data alone after gene X is knocked out makes it difficult to distinguish direct and indirect regulations. Blue nodes : genes with expression changes due to the KO. **b** Using time-series expression data after KO, in principle, can enable differentiation of direct and indirect regulations and infer regulation by non-KO genes. **c** RENGE infers a signed GRN from the time-series expression data after gene KO, obtained by scCRISPR analysis, by modeling the process in which the effects of the gene KO propagate on the network. **X** denotes the decrease in expression of the target gene due to KO (orange node), **AX** denotes the expression change due to direct regulation from the KO gene, and **A²X** denotes the expression change due to indirect regulation via another gene from the KO gene. $w(t, k, g)$ denotes the magnitude of the effect of the $k$-th order regulation from the KO gene $g$ at time $t$. **d** Comparison of methods on different features.

representing the network is calculated using the bootstrap method. RENGE represents the direct regulation from gene $j$ to gene $i$ by element $\{\mathbf{A}\}_{i,j}$, and $\mathbf{A}^2, \mathbf{A}^3, \cdots, \mathbf{A}^K$ capture the expression changes due to the indirect effects of higher-order regulation. Since higher-order regulation includes regulation by genes that were not knocked out, regulation by these non-KO genes can also be inferred.

**Benchmark using simulated data**. Time-series expression data after gene KO was used, thus information on gene regulation is included in both KO gene information (which gene was knocked out) and temporal information (what temporal expression changes occurred as a result). We compared RENGE with state-of-the-art methods (GENIE3, dynGENIE3, BINGO, MIMOSCA, scMAGeCK-LR/RRA) that also utilize this type of information. A comparison of the features for each method is presented in Fig. 1d. The proposed method, RENGE, makes full use of the

KO-gene and temporal information and can infer signed regulation (i.e., distinguish positive and negative regulation), including that by non-KO genes.

To compare these methods, we generated time-series scRNA-seq data after gene KO based on more than 600 GRNs using dyngen[17], which is a simulator of gene expression dynamics based on a regulatory network (see Methods for more detail). Network inference was then performed using each method. Agreement with the ground-truth network was evaluated using the area under the precision-recall curve (AUPRC) divided by that of a random predictor (AUPRC ratio). The AUPRC was calculated without self-regulation. First, we evaluated whether RENGE could correctly infer the presence or absence of regulation, ignoring the sign. We calculated the AUPRC ratio changing the ratio of the KO genes, which is the number of KO genes divided by the total number of genes in the network. Figure 2a, b, shows the $\log_2(\text{AUPRC ratio})$ averaged between GRNs for each backbone structure used to generate each GRN.

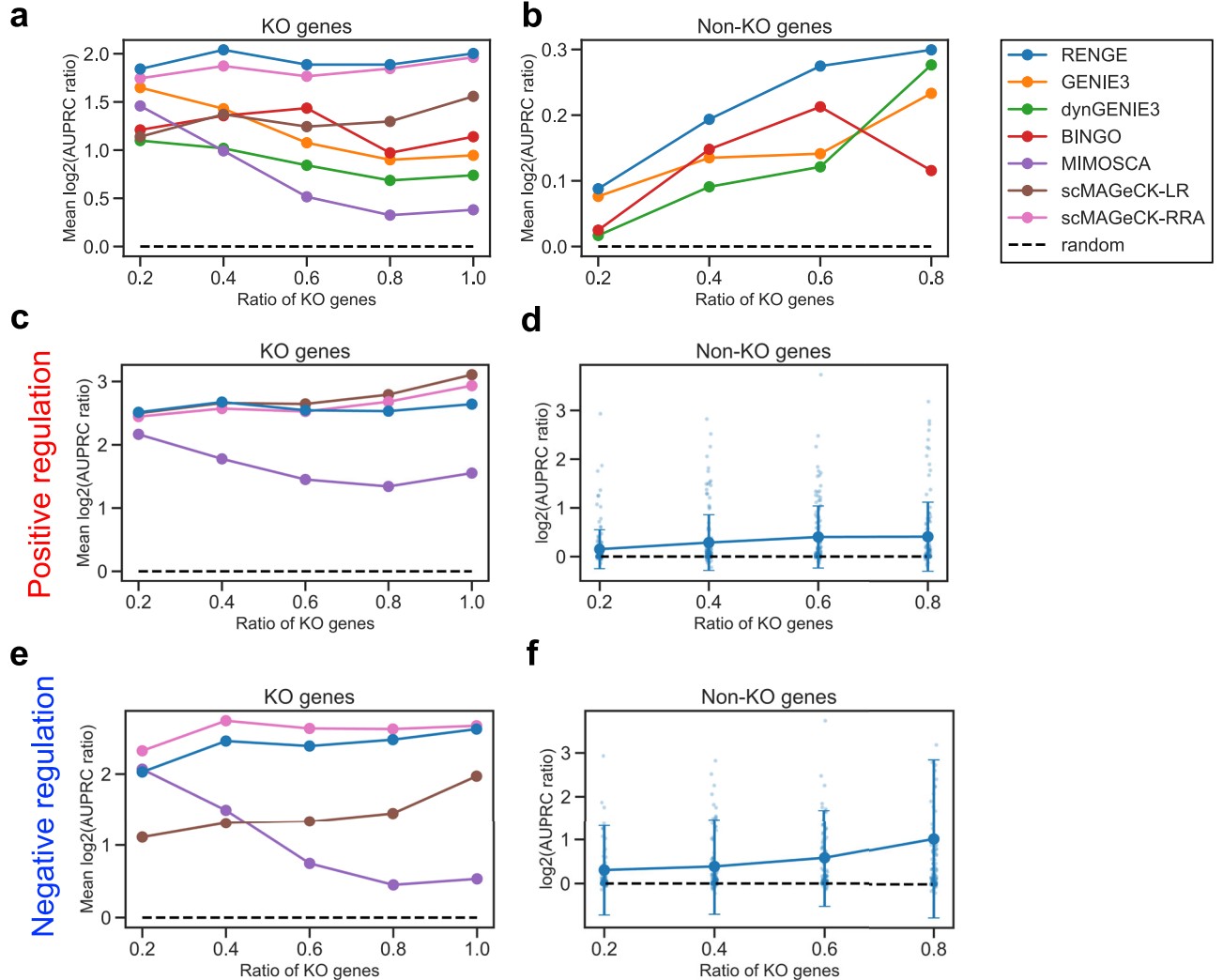

**Fig. 2 Benchmark results when using the simulated data set.** Horizontal axis: ratio of KO genes in the GRNs. $\log_2($ AUPRC ratio $)$ were averaged between three backbones. **a** Regulation by KO genes. **b** Regulation by non-KO genes. **c** Positive regulation by KO genes. **d** Positive regulation by non-KO genes. **e** Negative regulation by KO genes. **f** Negative regulation by non-KO genes. Error bars represent mean ± standard deviation.

The average AUPRC ratio for RENGE exceeded those of existing methods at various KO gene ratios (Fig. 2a, b). This trend was observed especially for regulation by non-KO genes. The greater the KO gene ratio, the higher the AUPRC ratio for regulation by non-KO genes, suggesting that knocking out more genes will provide more information on gene regulation. The results for GRNs for each of the three backbones are shown in Supplementary Figs. 1–3. Though RENGE showed superior performance on average, the performance of RENGE depended on the backbone used, which may be due to the complexity of the GRN. Next, we evaluated whether positive and negative regulations were correctly inferred. In inferring positive and negative regulation by KO genes, RENGE showed the competitive performance with the existing methods, respectively (Fig. 2c, e). In inferring positive and negative regulation by the non-KO genes, RENGE showed a higher AUPRC ratio than a random predictor (Fig. 2d, f). Of note, MIMOSCA and scMAGeCK-LR/RRA do not infer regulation by non-KO genes.

**scCRISPR analysis of hiPSCs**. To apply RENGE to an actual time-series scRNA-seq dataset, a scCRISPR experiment was performed that focused on the pluripotency network in human

iPSCs. It is well established that the core regulatory network that maintains pluripotency is composed of POU5F1, SOX2, NANOG, and PRDM14[18,19]. Assuming that approximately 5000 single cells can be captured in each sample, and to obtain approximately 100 cells per guide RNA (gRNA) via scRNA-seq, we limited the number of gRNAs to be analyzed in each experiment to 50 gRNAs. With two gRNAs per gene, a library can include 25 genes. We therefore selected 23 genes (TFs) that are thought to be involved in hiPSC pluripotency[18–20] and four control gRNAs (two AAVS1-targeting and two non-cutter gRNAs) to construct a gRNA library (Supplementary Table 1). To capture the time-series expression changes, samples were collected on days 2, 3, 4, and 5 after transduction and scRNA-seq analysis was performed (Fig. 3a). The number of gRNAs detected in each single cell was high on day 2, compared to the rest of the time points, which may have been caused by the proviral BFP expression from the provirus had not yet been saturated, resulting in cells with multi-copy proviral integrations being over-represented in the BFP+ fraction on day 2 (Supplementary Fig. 4). Although the number of cells assigned to each gRNA varied (Supplementary Fig. 5), an average coverage of 75 cells/gRNA was achieved, where cells bearing a single gRNA were counted. We then performed uniform manifold approximation

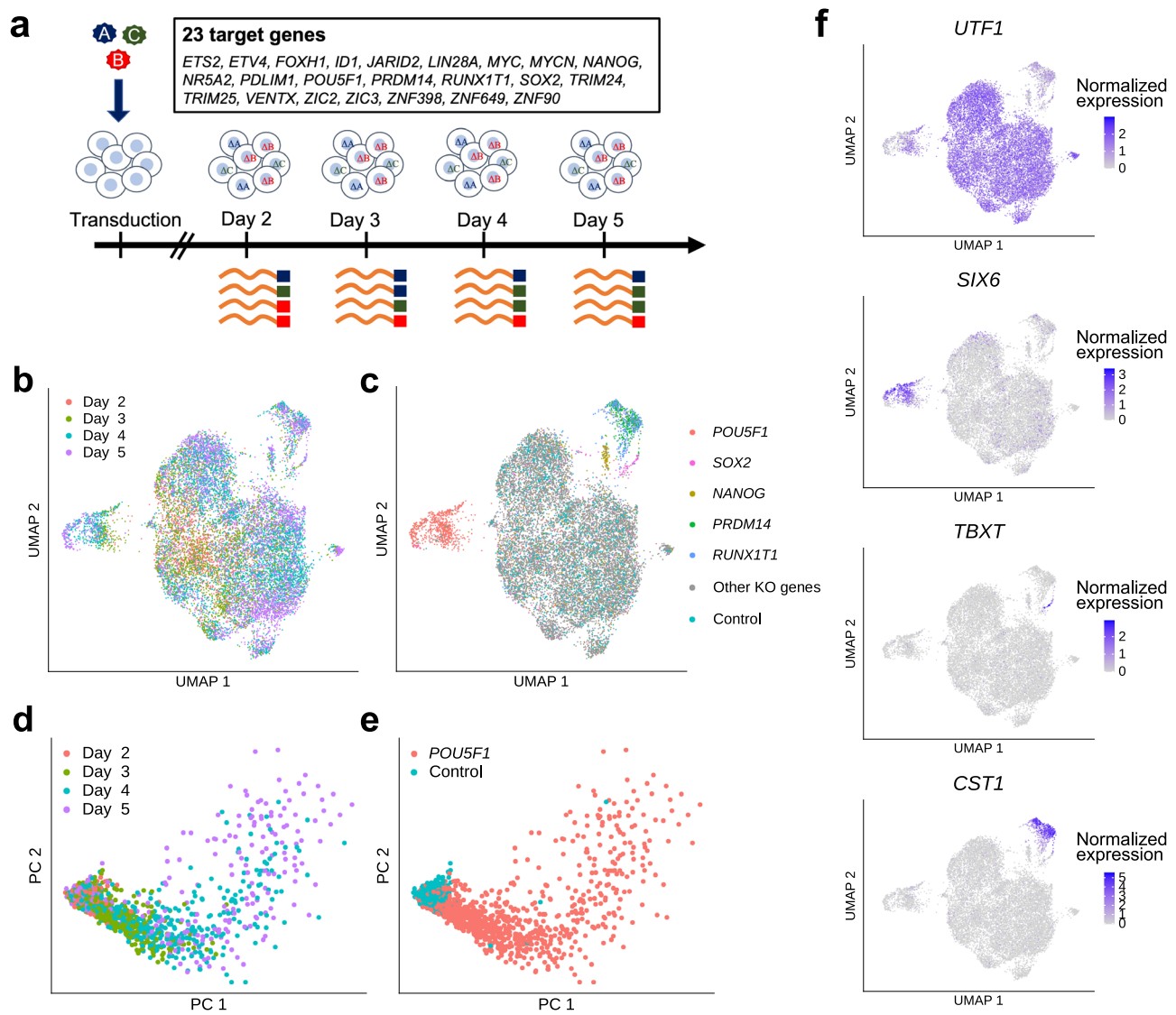

**Fig. 3 Time-series scCRISPR analysis of human iPS cells. a** Experimental design. scRNA-seq was performed on cells sampled 2, 3, 4, and 5 days after transduction by lentiviruses containing a gRNA vector. **b**, **c** UMAP plot for cells bearing a single gRNA. **d**, **e** PCA plot for cells bearing the gRNA of the control or *POU5F1*. Colors indicate the sampling day of the cells (**b**, **d**) or the target gene of a gRNA detected in a cell (**c**, **e**). **f** Marker expression including *UTF1* (pluripotency marker), *SIX6* (neural markers), *TBXT* (axial mesoderm marker), and *CST1* (definitive endoderm marker).

and projection (UMAP) on the cells with a single gRNA and observed a clear separation between the control cells and those with gRNAs targeting *POU5F1*, *SOX2*, *NANOG*, *PRDM14*, and *RUNX1T1* (Fig. 3b, c). Principal Component Analysis (PCA) of the control cells and *POU5F1* KO cells revealed that the KO influence on the transcriptome expanded over time (Fig. 3d, e). In each cluster on the UMAP plot, known markers are selectively expressed including *UTF1* (pluripotency marker) in control cells, *SIX6* (neural marker) in *POU5F1* KO cells, *TBXT* (axial mesoderm marker) in *SOX2* KO cells, and *CST1* (definitive endoderm marker), *FOXA2* and *EOMES* (mesoendoderm markers) in *PRDM14* KO and *RUNX1T1* KO cells (Fig. 3f, Supplementary Fig. 6). These results indicate that the time-series scCRISPR analysis sufficiently detects changes in gene expression after gene KO.

**Inferring network for pluripotency in human iPSCs.** Using the time-series scCRISPR analysis data for the hiPSCs, the seven methods were employed to infer the GRNs of 103 TFs, comprising the 23 KO genes and the top 80 TFs with the largest

expression changes due to gene KO. The expression changes were calculated using the coefficient matrix $\beta$ obtained by applying MIMOSCA. We constructed a subnetwork using ChIP-seq data of 19 genes in human pluripotent stem cells that were available in the ChIP-Atlas database[8] (Supplementary Table 3), and assumed the subnetwork as a ground truth. The correctness of the part of inferred 103-gene networks was evaluated based on their agreement with the corresponding ground-truth subnetwork, which is represented by the AUPRC ratio. The ChIP-seq data contained the score ($-10 \times \log_{10}$(MACS2 $q$-value)) to represent the significance of the TF binding to DNA; only bindings with a score higher than the threshold (ChIP threshold) were considered as regulatory in the ground-truth network. Therefore, the ground-truth network changes depending on the ChIP threshold. That is, the higher the ChIP threshold, the more reliable the regulations in the obtained network. The AUPRC ratio was calculated as a function of the ChIP threshold. For most thresholds and regulations by the KO and non-KO genes, RENGE had the highest AUPRC ratios among the methods used (Fig. 4a, b). For RENGE, the AUPRC ratio increased as the ChIP threshold increased for

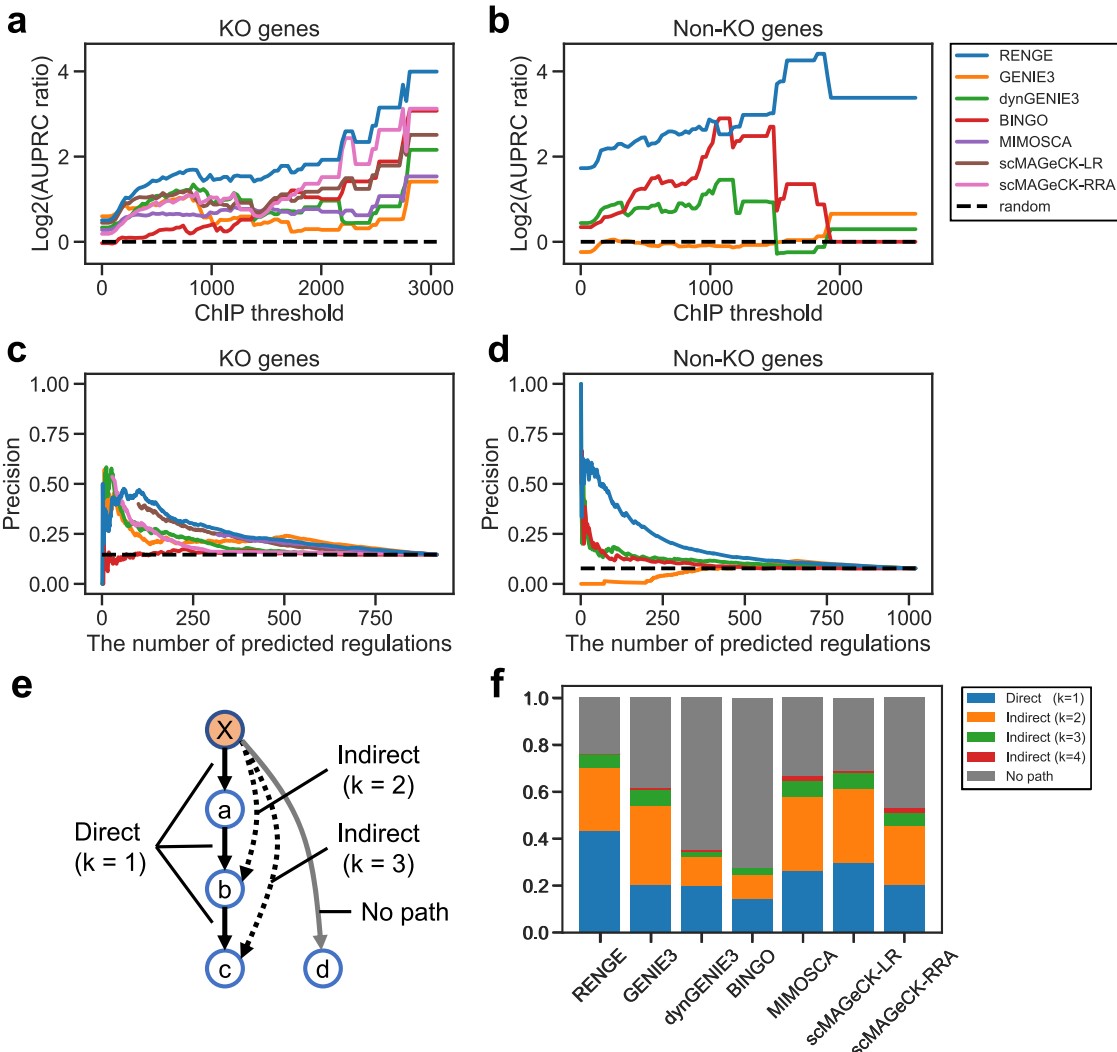

**Fig. 4 Benchmark results for the hiPSC data. a** AUPRC ratios for regulation by KO genes. Horizontal axis: threshold of the confidence level ($-10 \times \log_{10}$(MACS2 $q$-value)) for the TF binding in ChIP-seq data. **b** AUPRC ratios for regulation by non-KO genes. **c, d** Precision comparison of the methods. Horizontal axis: number of inferred regulations corresponding to a threshold of a significance score such as $p$-value. **c** Regulations by KO genes. **d** Regulations by non-KO genes. **e** Schematic diagram showing the direct, indirect, and no path regulations. Solid black arrows: regulation of the ground-truth network. If these regulations are inferred, they are called direct. If the dashed arrows are inferred, they are called indirect regulation via the $k-1$ ($k \geq 2$) genes. If the gray arrow is inferred, it is called no path, which represents regulation that is neither direct nor indirect. **f** Classification of regulations inferred by each method. The ground-truth network was constructed with ChIP threshold $= 300$.

regulation by KO genes and non-KO genes, while for BINGO, GENIE3, and dynGENIE3, the AUPRC ratio decreased when the ChIP threshold was high for regulation by non-KO genes. These results imply a higher correlation than compared methods between the $q$-values of the regulations inferred by RENGE and those of the ChIP-seq data. We then calculated the correlation coefficients for the confidence level of the regulation calculated by each method and the ChIP-seq ($-10 \times \log_{10}$(MACS2 $q$-value)) (Supplementary Table 4). The confidence of regulation inferred by RENGE was expressed as $-\log_{10}(q\text{-value})$. Indeed, for regulation by non-KO genes, the correlation coefficient for RENGE (0.290) was much higher than those for other methods; however, that for KO gene regulation was highest for GENIE3 (0.390) followed by RENGE (0.292). These results suggest that RENGE can infer more reliable regulations than the other methods, particularly for non-KO genes.

Thus far, we have regarded the binding of TFs to DNA detected by ChIP-seq as representing ground-truth regulation.

However, considering that TF binding may not necessarily mean regulation, i.e., false negatives in a GRN inferred by RENGE can be acceptable, we also compared the methods focusing only on the proportion of the inferred regulations that were supported by TF binding, i.e., precision. The ground-truth network was constructed with a ChIP threshold $= 300$. Consistent with the previous analysis, RENGE showed higher precision than the other methods, particularly in regulation by non-KO genes (Fig. 4c, d).

The details of the inferred regulations by each method were then examined by comparing them with the ground-truth network with a ChIP threshold $= 300$. For each method, 237 inferred regulations, which is the number of regulations in the ground-truth network, were extracted in order of confidence score and classified as follows. An inferred regulation from gene $j$ to gene $i$ was classified as direct when the shortest path length ($k$) from gene $j$ to gene $i$ in the ground-truth network was 1, and indirect when $k > 1$. If there is no path from gene $j$ to gene $i$ in the ground-truth network, the inferred regulation is classified as no

path (Fig. 4e). RENGE had the highest percentage (43%) of direct regulation and the lowest (23%) of no path regulation (Fig. 4f). Meanwhile, dynGENIE3 and BINGO had a high percentage of no path regulation, likely because they did not fully utilize the information on the KO genes, but rather primarily inferred regulation based on temporal changes in expression. In GENIE3, scMAGeCK-LR/RRA, and MIMOSCA, the percentage of no path regulation was low, while that of indirect regulation was high. Hence, these methods may not effectively distinguish between direct and indirect regulation as they do not incorporate temporal information.

**Features of pluripotency network inferred by RENGE.** We focused on the regulations with an FDR < 0.01 that were inferred by RENGE (Fig. 5a). The number of positive and negative regulations from each gene are shown in decreasing order in Fig. 5b. The top 20 genes out of 103 genes (19%) ordered by the number of out-edges are associated with 51% of the regulations detected, suggesting that these genes are important for maintaining pluripotency. These 20 genes were extracted and shown in Fig. 5c, and included pluripotency factors, such as POU5F1, NANOG, SOX2, and PRDM14. We also investigated the structure of the subnetwork comprising these 20 genes (Fig. 5d). The pluripotency factors form a positive feedback loop, which is considered important for the maintenance of pluripotency[18,19]. Moreover, negative regulations from the pluripotency factors to CHD7 and CTNNB1 were detected, which were also supported by ChIP-seq data. The expression of these genes is reportedly important for cell differentiation[21,22]. Collectively, these findings imply that the pluripotency factors may suppress differentiation by directly inhibiting the expression of the differentiation-related genes. Although the pluripotency factors also positively regulate UTF1, but the regulation from UTF1 to the pluripotency factors was only weakly suggested by RENGE (Fig. 5e, f), which is consistent with reports that have indicated that UTF1 expression can serve as a marker of pluripotency, however, is not responsible for maintaining pluripotency[23]. Thus, the structure of the inferred network, including the sign of each regulation, is considered reliable, based on comparison with the known characteristics of pluripotency gene regulations.

From the inferred GRN, we found that certain gene pairs have a similar set of target genes (Supplementary Fig. 7), which may form a protein complex in transcriptional regulation. We validated this hypothesis using the protein-protein interaction data obtained from the STRING database[24] and protein complex data obtained from the CORUM3.0 database[25]. To quantify the similarity of the target genes between a gene pair, we calculated the regulatory correlations between each gene pair as a correlation between the regulatory coefficients of the genes. The gene pairs with large positive regulatory correlations (i.e. the same target genes regulated in the same direction) determined by RENGE tended to have high STRING scores and be included in the CORUM complex (Fig. 6a). We also calculated regulatory correlation using the other GRN inference methods (Supplementary Figs. 8, 9); those calculated using GENIE3 also showed the differences between all gene pairs and gene pairs in the CORUM complex or with a high STRING score, however, one calculated using dynGENIE3 and BINGO showed no clear difference between the groups (Supplementary Fig. 8). Although limited to the gene pairs among the KO genes, MIMOSCA and scMAGeCK-LR also showed a clear difference (Supplementary Fig. 9). These results suggest that regulatory correlations may reflect protein complex formation even if the regulatory correlations are calculated without distinguishing direct and indirect regulation.

We focused on the gene pairs with positive STRING scores that also had high regulatory correlations, which may function as a complex (Fig. 6b). It has been suggested that the chromatin remodeler CHD7 and the DNA topoisomerase TOP1 (correlation = 0.90) can physically interact and are involved in the transcription of long genes in the neuron[26]. Though the distribution of the co-target gene length regulated by both CHD7 and TOP1 was similar to that of all genes in the GRN, the co-target genes included *JARID2* (Supplementary Fig. 10). This 276 kb gene is expressed in neurons and its mutations and deletions cause neurodevelopmental syndrome[27]. Regarding the pluripotency core factor POU5F1 and histone lysine demethylase JMJD1C (correlation = 0.83), it is suggested that POU5F1 recruits JMJD1C near POU5F1 target genes and prevents DNA methylation by DMNT3A through histone demethylation by JMJD1C[28]. For CTNNB1 and JADE1 (correlation = −0.91), it is known that JADE1 binds to, and ubiquitinates, CTNNB1[29], which is consistent with a negative regulatory correlation (Fig. 6c). Meanwhile, the regulatory correlation between PRDM14 and RUNX1T1 (0.92) was the highest of any gene pair (Fig. 6b). In mice, PRDM14 and RUNX1T1 have been suggested to form a complex by shotgun liquid chromatography-tandem mass spectrometry (LC-MS/MS)[30]. Thus, the PRDM14 and RUNX1T1 complex may be a key factor in the maintenance of pluripotency (Fig. 6c).

When gene pairs with large absolute values for the regulatory correlations function as complexes, we expect that their genomic binding positions may be co-localized. To test this hypothesis, we obtained colocalization data for TF binding from the ChIP-Atlas for each of the 19 genes with available ChIP-seq data in pluripotent stem cells. There was a weak but statistically significant correlation (correlation = 0.28, *p*-value < 0.0005) between the absolute regulatory correlation of a gene pair and the co-localization score of the genomic binding position of the pair (Fig. 6d). The top 3 gene pairs with the highest colocalization score were (NANOG, POU5F1), (NANOG, CHD7), and (NANOG, SOX2). (NANOG, POU5F1) and (NANOG, SOX2) also showed relatively high STRING scores. NANOG, POU5F1, and SOX2 are the core pluripotency factors and are known to function together[18,19]. NANOG and CHD7 are reported to co-bind enhancers and have the opposite effect on the expression of target genes in mice[31]. Note, however, that large absolute regulatory correlation and STRING scores do not necessarily indicate colocalization of genomic binding positions. Indeed, there are cases where a pair of genes regulates common targets via mechanisms independent of genome co-binding. For example, CTNNB1 and JADE1 have a large absolute regulatory correlation due to ubiquitination. Thus, the signed regulatory network inferred by RENGE can be used to estimate the interactions of multiple genes in transcriptional regulation.

**Prediction of expression changes induced by gene knockout.** Based on the GRN inferred using RENGE, we then examined whether RENGE could predict changes in the expression of each gene after knocking out any gene in the GRN. A total of 103 genes with 80 non-KO genes and 23 KO genes constituted the node set for the focal system. To this end, we split the hiPSC data into training data and test data. The test data comprised expression data after one of the 23 KO genes was knocked out, while the training data comprised expression data for the other KO genes. The RENGE model was trained using the training data and we examined whether the RENGE model could predict the expression changes in the test data. For instance, the RENGE model trained without the *NANOG* KO data was able

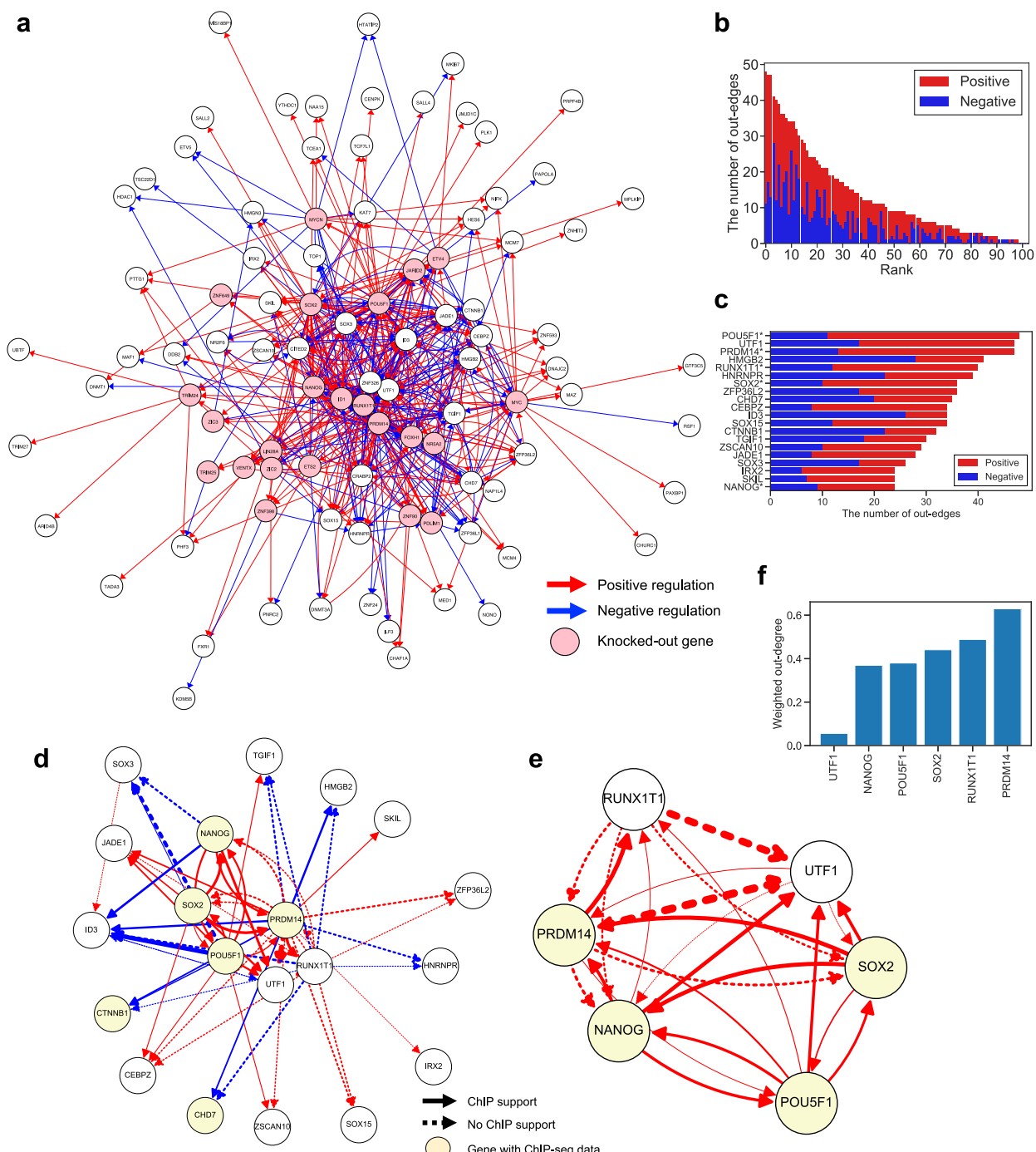

**Fig. 5 hiPSC regulatory network inferred by RENGE. a** GRN of hiPSCs. Regulation was detected with FDR < 0.01. Only the edges of the top 35% absolute regulatory coefficients and the nodes with at least one edge are shown (474 edges and 95 nodes). Red arrows: positive regulation, blue arrows: negative regulation. Arrow thickness indicates the magnitude of the regulatory coefficient. Pink nodes: KO genes. **b** Number of positive and negative regulations from each gene. **c** The top 20 genes with the highest out-degree extracted from (**b**). The genes with asterisks indicate KO genes. **d** GRN of the top 20 genes with the highest out-degree. Only the edges of the top 20% absolute regulatory coefficients are shown. Solid arrows: regulations supported by ChIP-seq data ($-10 \times \log_{10}$(MACS2 $q$-value) > 50) obtained from ChIP-Atlas, and dotted arrows: regulations not supported. **e** The subnetwork at the center of the GRN in (**d**). All edges with FDR < 0.01 are shown. **f** Weighted out-degree of each gene in the GRN in (**e**). Weighted out-degree was calculated as the sum of absolute regulatory coefficients from each gene.

to accurately predict (correlation = 0.70, $p$-value < $10^{-15}$) changes in the expression of the other 102 genes in the network on day 5 after *NANOG* KO (Supplementary Fig. 11a). Overall, RENGE could predict the expression changes with an average correlation coefficient of approximately 0.3 (Supplementary Fig. 11b). The correlation coefficients for each KO gene and

each day are shown in Supplementary Table 5. On day 2, the correlation coefficients were higher overall, suggesting that changes in the expression of genes directly regulated by the KO genes, which already occurred on day 2, were easier to predict, while changes in downstream gene expression, which occurred later, were harder to predict.

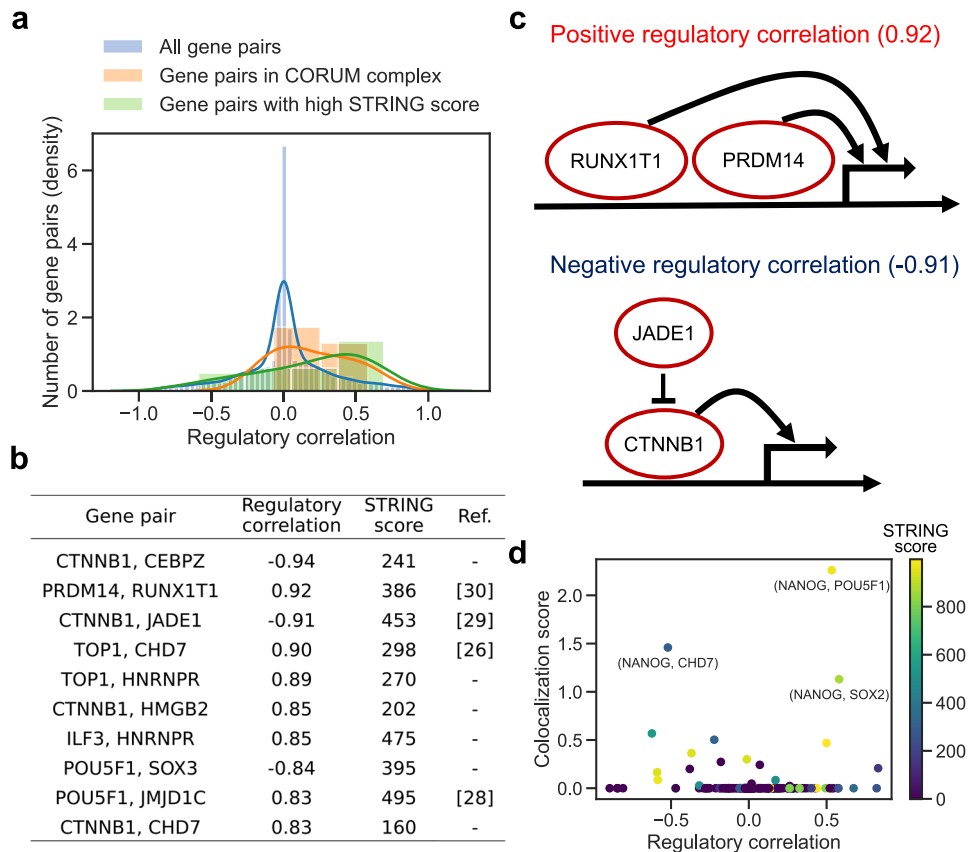

**Fig. 6 Relationship between regulatory correlations and protein complex formation.** Regulatory correlations are calculated as correlations between the regulatory-coefficient vectors for each gene pair in the network. A positive regulatory correlation indicates that the same genes are regulated in the same direction by the pair, while a negative regulatory correlation indicates that the same genes are regulated in the opposite direction. **a** Distribution of the regulatory correlations among all possible gene pairs versus those in the genes in protein complexes obtained from CORUM3.0 versus those among gene pairs with a high STRING score. The STRING score represents the confidence level of the protein-protein interactions. **b** Top 10 gene pairs with the highest absolute regulatory correlation among gene pairs with a positive STRING score. **c** Examples of gene pairs with large negative and positive regulatory correlations. **d** Relationship between regulatory correlations and colocalization scores of TF binding. Colocalization was calculated from ChIP-seq data in pluripotent stem cells by the ChIP-Atlas. Colors indicate STRING scores. Gene names for the top 3 pairs with the highest colocalization score are shown.

## Discussion

Although many studies have previously inferred GRNs from expression changes following gene KOs, most have done so by using snapshots of these changes[9,32]. Moreover, while scCRISPR analysis has enabled the exhaustive KO of genes and measurement of outcomes using scRNA-seq, to date, it is primarily employed to obtain a snapshot of gene expression after KO[13,15,16]. As such, GRN inference methods based on these snapshot data fail to effectively infer direct causality as influence of KO expands to genes that are not directly regulated by KO genes. Meanwhile, RENGE is a GRN inference method that overcomes this limitation by using temporal changes in gene expression following KO, while also effectively inferring regulation by non-KO genes.

The performance of RENGE was validated using simulated data and hiPSC data, both of which highlighted its ability to infer networks more accurately than existing methods. In fact, the pluripotency gene regulatory network inferred by RENGE from the transcriptome data obtained by a single experiment had a high level of consistency with various findings revealed by the actual data that had been accumulated over many years. However, the inferred network also identified a PRDM14 and RUNX1T1 complex as a previously unknown key factor for pluripotency maintenance.

Nevertheless, certain precautions should be considered when using RENGE based on the implicit assumptions of the model.

First, RENGE assumes that each gene regulates its target gene independently. Therefore, the cooperative effects between multiple regulator genes cannot be considered. Second, the analysis in this study only dealt with expression data from cells with a single gene knocked out. However, scCRISPR analysis also provides expression data for cells in which multiple genes are simultaneously knocked out. Data from multiple KOs may be utilized to model the cooperative effects of multiple regulators. Third, the RENGE model also assumes that the time evolution of gene expression is discrete and simultaneous in all genes, thus, regulations that are extremely fast or slow compared to others may not be appropriately inferred. Hence, the sampling interval of the cells is also important; if the time evolution of gene expression in a real system is notably faster than the sampling rate, a portion of the direct regulations may be overlooked. Therefore, it is desirable to sample cells in a manner that ensures that the time evolution of the focal system progresses only one or two steps with every sampling interval. In addition, we note that the computational time and memory usage for RENGE may vary with the size of GRN and the dataset used. A comprehensive benchmarking analysis to quantify these dependencies has not been performed in this study, and it is possible that for certain datasets, RENGE may require more computational time compared to the existing methods.

Since RENGE can predict the changes in gene expression induced by a gene KO, it could be used for step-by-step

determination of a GRN by repeating perturbation experiments and network inference by RENGE. That is, first, a perturbation experiment will be performed. Second, a network will be inferred by RENGE. Third, based on the inferred network, the genes to be knocked out in the next experiment will be determined to more accurately elucidate the GRN. For example, if a gene KO is predicted to have a large influence on other genes by RENGE, then it may be the next to be knocked out in subsequent experiments.

Using GRN more accurately inferred by RENGE, it may be possible to identify the genes responsible for governing the system dynamics. For example, a set (feedback vertex set; FVS) of genes determined solely from the structure of the GRN, was proven to be a set of key factors that could be used to observe/control the dynamics of the entire system[33,34]. In fact, it has been reported that by experimentally manipulating six factors included in the FVS, identified from the GRN structure of ascidians, each of the seven tissues (epidermis, brain, nerve cord, muscle, notochord, mesenchyme, and endoderm) can be specifically induced[5,35]. Similarly, it may be possible to discover TFs that are important for inducing hiPSC differentiation based on the GRN inferred by RENGE. Additional possible future extensions for RENGE are discussed in Supplementary Note 2.

## Methods

**Cell culture**. Human iPSC line, OILG-3, was obtained from the Wellcome Sanger Institute and cultured in StemFlex medium (Thermo Fisher) on Vitronectin (Thermo Fisher)-coated culture dishes. Cells were detached using TrypLE (Thermo Fisher) and re-seeded at $4 \times 10^4$ cells per well into 6-well plates for routine maintenance. For the first 24 h after passaging, cells were treated with 10 $\mu$M Y-27632 (Wako). SpCas9-expressing OILG cells were generated as previously described[36].

**gRNA cloning, lentiviral transduction, and single-cell RNA-seq**. Selected gRNAs (Supplementary Table 1) were cloned into pKLV2-U6gRNA5(BbsI)-PGKpuroBFP-W. Lentivirus was produced individually by transfecting 293FT cells together with lentiviral packaging plasmids, psPAX2 and pMD2.G using LipofectamineLTX[37]. The resulting viral supernatants were then pooled in an equal volume ratio. OILG-Cas9 ($1.5–6 \times 10^5$) cells were transduced with the pooled lentivirus at 8–9% transduction efficiency and maintained until harvesting without passaging. On days 2, 3, 4, and 5 after transduction, $8 \times 10^4$ BFP+ cells were collected using an MA900 cell sorter (Sony), then resuspended at $1 \times 10^6$ cells/mL in 0.05% BSA in PBS. These cells were then subjected to 5' scRNA-seq library preparation using a Chromium Next GEM Single Cell 5' Library & Gel Bead Kit following the manufacturers' protocol with minor modifications to simultaneously capture guide RNA molecules. Briefly, a spike-in oligo (5'-AAGCAGTGGTATCAACGCAGAGTACCAAGTTGATAA CGGACTAGCC-3') was added to the reverse transcription reaction. The 'small DNA' fraction isolated after cDNA clean-up was then used to generate a gRNA sequencing library with the primers listed in Supplementary Table 2. PCR was performed using $2 \times$ KAPA Hi-Fi Master Mix with the following program: 95 °C for 3 min, 12 cycles of 98 °C for 15 sec and 65 °C for 10 sec, followed by 72 °C for 1 min. The resulting gene expression libraries and gRNA libraries were pooled at a molecular ratio of 7:1 and sequenced using NovaSeq with 26 cycles for read 1, 91 cycles for read 2, and 8 cycles for the sample index.

**Alignment, gRNA assignment, filtering, and normalization**. A digital expression matrix with gRNA assignment was obtained using the CRISPR Guide Capture Analysis pipeline of Cell Ranger 5.0.0 (10x Genomics). The generated expression matrix was processed using Seurat (version 4.0.3)[38]. Single cells were filtered to leave cells with > 200 and < 10000 expressed genes and < 20% reads from mitochondrial genes. The expressions were normalized using the sctransform method of Seurat. Only cells bearing a single gRNA were used for downstream analysis.

**Modeling expression dynamics following gene knockout**. We investigated GRNs whose nodes were TFs only. Below, we adopt a 1-origin indexing system for all vectors and matrices. Consider a model that represents the propagation of the KO effect from the KO gene $g$ on the GRN. Let $G$ denote the number of genes included in the GRN. The $G$-dimensional gene expression vector $\mathbf{E}'_{g,K'}$ of a cell including the up to $K'$-th order regulatory effect from the KO gene $g$ is modeled as follows:

$$\mathbf{E}'_{g,K'} = \sum_{k'=1}^{K'} \left( \mathbf{M}_g \odot \mathbf{A} \right)^{k'} \mathbf{X}_g + \mathbf{b}_{K'}, \quad (3)$$

where $\mathbf{X}_g$ is a $G$-dimensional vector of which $g$th component is the expression change of gene $g$ due to its KO, and the other components are zero. When the cell is the wild type, i.e. no gene is knocked out ($g = 0$), $\mathbf{X}_0$ is a zero vector. $\mathbf{b}_{K'}$ is the $G$-dimensional expression vector corresponding to the wild type. $\mathbf{A}$ is a $G \times G$ matrix and $\mathbf{A}_{i,j}$ ($i \neq j$) represents the strength of regulation from gene $j$ to $i$; that is, the change in gene $i$ expression due to a unit amount change in gene $j$ expression. $\mathbf{A}_{i,j}$ ($i = j$) represents effects such as degradation and self-regulation (Supplementary Note 1). $\odot$ denotes an element-wise product. Eq. (3) is an extension of Eq. (1) with a mask matrix $\mathbf{M}_g$ representing that the KO gene $g$ is no longer regulated by other genes:

$$\{\mathbf{M}_g\}_{i,j} = \begin{cases} 0 & (i = g) \\ 1 & (i \neq g). \end{cases} \quad (4)$$

Thus, $\sum_{k'=1}^{K'} (\mathbf{M}_g \odot \mathbf{A})^{k'} \mathbf{X}_g$ represents the expression change from the wild type due to gene KO.

From the scCRISPR analysis, we obtained the $G$-dimensional gene expression vector $\mathbf{E}_{c,t}$ in cell $c$ sampled at time $t$ and $G$-dimensional vector $\mathbf{X}_{c,t}$ representing the decrease in expression of the KO gene in the cell ($t = 1, \cdots, T, c = 1, \cdots, C_t$). Here, $T$ is the number of time points, and $C_t$ is the number of cells sampled at time $t$. Note that here, in contrast to Eq. (2) in the Results section, the subscript of $\mathbf{E}$ have been changed from $g, t$ to $c, t$. The KO gene in cell $c$ sampled at time $t$ is identified by the presence of gRNA and denoted by $g_{c,t}$. The calculation of $\mathbf{X}_{c,t}$ from $g_{c,t}$ will be explained in a later section.

Suppose we have the gene expression data $\mathbf{E}'_{g,K'}$ ($K' = 1, \cdots, max\_K'$), in which the effects of different maximum orders of $K'$ regulation appear, we can infer the GRN $\mathbf{A}$ by fitting Eq. (3) to the data. However, it is impossible to synchronize the sampling time $t$ of the cells and the time at which the effects appear for up to the $K'$-th order of regulation from the KO gene. Hence, the maximum order of regulation from the KO gene in the cells at sampling time $t$ is unknown. Thus, RENGE estimates the value from the data. By introducing a term $w(t, k, g_{c,t})$ representing the strength of the effect of the $k$-th order of regulation at time $t$ when the gene $g_{c,t}$ is knocked out, we can express Eq. (3) as follows:

$$\mathbf{E}_{c,t} = \sum_{k=1}^{K} w(t, k, g_{c,t})(\mathbf{M}_{c,t} \odot \mathbf{A})^k \mathbf{X}_{c,t} + \mathbf{b}_t \quad (5)$$

$$w(t, k, g_{c,t}) = \frac{1}{1 + \exp^{-(\alpha_{g_{c,t}} + \beta t - \gamma k)}}, \quad (6)$$

where $w(t, k, g_{c,t})$ is assumed to be monotonically increasing with respect to $t$ and monotonically decreasing with respect to $k$, thus,

as time progresses, the effects of higher-order regulation become more apparent. $\alpha_{g_{c,t}}$, $\beta$, $\gamma$ are the parameters to be estimated, and $\beta \geq 0$, $\gamma \geq 0$. The parameter $\alpha_{g_{c,t}}$ represents the time required for the effect of the KO of gene $g_{c,t}$ to appear and is assumed to differ with each KO gene. $\beta$ is related to a rate constant at which the regulation step progresses with respect to time $t$, and $\gamma$ is a parameter representing the degree of decrease in the effect of higher-order regulation. $\mathbf{M}_{c,t}$ is obtained by replacing the subscripts of the mask matrix in Eq. (4) with the relation $g = g_{c,t}$. The parameters to estimate are $\mathbf{A}$, $\mathbf{b}_t$ ($t = 1, \cdots, T$), $\alpha_{g_{c,t}}$ ($g_{c,t} = 1, \cdots, G_{ko}$), $\beta$, $\gamma$, where $G_{ko}$ is the number of KO genes.

**Parameter estimation.** The parameters are estimated by minimizing the following objective function:

$$L = \sum_{t=1}^{T} \sum_{c=1}^{C_t} \left\| \mathbf{m}_{c,t} \odot \left[ \mathbf{E}_{c,t} - \left\{ \sum_{k=1}^{K} w(t, k, g_{c,t})(\mathbf{M}_{c,t} \odot \mathbf{A})^k \mathbf{X}_{c,t} + \mathbf{b}_t \right\} \right] \right\|_2^2$$
$$+ \lambda_1 \sum_{i,j=1}^{G} \left| \{\mathbf{A}\}_{i,j} \right| + \lambda_2 \sum_{k=1}^{K} \sum_{i,j=1}^{G} \{\mathbf{A}^k\}_{i,j}^2, \quad (7)$$

where $\{\mathbf{A}\}_{i,j}$ denotes the $i, j$ element of the matrix $\mathbf{A}$, $\odot$ denotes the element-wise product, and $\mathbf{m}_{c,t}$ is the mask vector for cell $c$ at time $t$:

$$\{\mathbf{m}_{c,t}\}_i = \begin{cases} 0 & (i = g_{c,t}) \\ 1 & (i \neq g_{c,t}) \end{cases}. \quad (8)$$

The first term in Eq. (7) is the squared error between the predictions of the model and the data. $\mathbf{m}_{c,t}$ is used to ignore the squared error of KO gene $g_{c,t}$ expression in cell $c$ at time $t$ because mRNA of KO gene $g_{c,t}$ may still be expressed even when the functional protein is lost when using the CRISPR system. The last two terms in Eq. (7) are the L1 and L2 regularization terms of the parameter $\mathbf{A}$, respectively. To suppress the magnitude of each element of not only $\mathbf{A}$ but also $\mathbf{A}^k$ ($k \geq 2$), an L2 regularization term was added for $\mathbf{A}^k$ ($k = 1, \cdots K$). Note that the L1 regularization term was only added for $\mathbf{A}$ and not for $\mathbf{A}^k$ ($k \geq 2$) because $\mathbf{A}$ represents a GRN and thus is expected to be sparse, but $\mathbf{A}^k$ ($k \geq 2$) is not necessarily sparse. The objective function is minimized using the L-BFGS-B method implemented in scipy.minimize. $K, \lambda_1, \lambda_2$ are hyperparameters that are set to values that minimize cross-validation loss using Bayesian optimization with Optuna[39].

**Calculation of $\mathbf{X}_{c,t}$.** One of the RENGE inputs, $\mathbf{X}_{c,t}$, is a $G$-dimensional vector representing the decrease in expression of the target gene due to its KO in cell $c$ at time $t$. Here, we assumed that when the target gene is entirely knocked out, the gene expression is decreased to zero. That is, the decrease in expression equals the average expression in control cells. However, in scCRISPR analysis, the target gene is not necessarily knocked out even in cells where the corresponding gRNA is detected. It is therefore necessary to distinguish between cells in which the transcriptome is affected by the KO and cells in which the KO fails and thus the transcriptome is not affected. RENGE uses the concept of perturbation probability, defined as the probability that gRNA detected in a cell has an effect on the transcriptome. RENGE calculates the perturbation probability $p_c$ ($c = 1, \cdots, C$) for each cell $c$ in the same way as MIMOSCA[13], where $C$ is the total number of cells.

$\mathbf{X}_{c,t}$ is defined as the decreased expression of the KO gene $g_{c,t}$ multiplied by $p_c$:

$$\mathbf{X}_{c,t,i} = \begin{cases} -p_c \cdot \frac{1}{C_t^{ctrl}} \sum_{j=1}^{C_t^{ctrl}} \mathbf{E}_{j,t,i}^{ctrl} & (i = g_{c,t}) \\ 0 & (i \neq g_{c,t}), \end{cases} \quad (9)$$

where $C_t^{ctrl}$ is the number of control cells at time $t$ and $\mathbf{E}_{j,t,i}^{ctrl}$ is the expression of gene $i$ in control cell $j$ at time $t$.

**$p$-value calculation using the bootstrap method.** RENGE calculates the $p$-value for each element of the matrix $\mathbf{A}$, which indicates the strength of regulation, using the bootstrap method as follows. Let the data set be denoted by $\mathbf{D} = \bigcup_{t=1}^{4}(\mathbf{X}_t, \mathbf{E}_t)$. The bootstrap data set $\mathbf{D}_1, \cdots, \mathbf{D}_N$ is created by sampling cells with replacement, keeping the number of cells for each KO gene at each time point ($N = 30$ by default). For each $\mathbf{D}_l$ ($l = 1, \cdots, N$), apply RENGE and estimate $\mathbf{A}_l$. Given $\mathbf{A}_l$ ($l = 1, \cdots, N$), calculate the sample variance $Var(\{\mathbf{A}\}_{i,j})$ ($i, j = 1, \cdots, G$) of $\{\mathbf{A}\}_{i,j}$. Assuming the null distribution of $\{\mathbf{A}\}_{i,j}$ is $\mathcal{N}(0, Var(\{\mathbf{A}\}_{i,j}))$, RENGE calculates the $p$-value $p_{i,j}$ of $\{\mathbf{A}\}_{i,j}$ as follows:

$$p_{i,j} = \begin{cases} 2(1 - \Phi^{-1}(\{\mathbf{A}\}_{i,j}/Var(\{\mathbf{A}\}_{i,j}))) & (\{\mathbf{A}\}_{i,j} \geq 0) \\ 2(\Phi^{-1}(\{\mathbf{A}\}_{i,j}/Var(\{\mathbf{A}\}_{i,j}))) & (\{\mathbf{A}\}_{i,j} < 0), \end{cases} \quad (10)$$

where $\Phi$ is the cumulative distribution function of the standard normal distribution. The $q$-value is then calculated using the Benjamini-Hochberg procedure to control for multiple hypothesis testing. Since RENGE cannot infer self-regulation, all downstream analyses, including method comparison and network analysis, were performed by excluding self-regulation.

**Other GRN inference methods.** The following existing methods were compared with RENGE: GENIE3[9], dynGENIE3[40], BINGO[32], MIMOSCA[13], and scMAGeCK[16]. GENIE3 predicts the expression of a gene from that of other genes using a tree-based ensemble. The importance of one gene for the prediction of another indicates the strength of the interaction between the genes. Although it exhibited superior performance in the benchmark of GRN inference from scRNA-seq data[11], GENIE3 cannot handle information on KO genes or time series data. In this study, one cell was treated as one sample, and time information was ignored. In each cell, the expression of the target KO gene was set to 0 regardless of its measured mRNA expression.

dynGENIE3 is a modified version of GENIE3 that is appropriate for time-series data; however, it cannot handle KO gene information. In this study, at each time point, the expression of each cell for each KO gene was averaged to produce a time series data set of (number of KO genes +1). In each time-series data set, the expression of the KO gene was set to 0.

BINGO is a method used to infer GRNs from time-series expression data by modeling gene expression dynamics with stochastic differential equations involving nonlinear gene-gene interactions. It can also handle KO information. BINGO takes two types of input data, time-series expression data (as data.ts) and KO gene data (as data.ko). The time-series data was constructed in the same way as for dynGENIE3, and KO gene data was constructed based on gRNA assignment.

MIMOSCA was developed for scCRISPR-screening data, and performs a linear regression of expression data using the gRNA detected in each cell and other information as covariates. This method can handle the index of the time point from which each cell is derived as a covariate, but not the time-series information. In this study, we used MIMOSCA by setting gRNA and the index of timepoint as covariates.

scMAGeCK includes scMAGeCK-LR and scMAGeCK-RRA, both GRN inference methods for the scCRISPR-screening data. scMAGeCK-LR performs linear regression similar to MIMOSCA. scMAGeCK-RRA uses Robust Rank Aggregation (RRA) to detect genes with expression changes in each KO. However, it cannot handle time information, so we applied scMAGeCK by ignoring the time information of each cell.

Recently, SCEPTRE[41] and Normalisr[42] were shown to improve the inference of associations between perturbations and gene expression in scCRISPR analysis. However, since these methods were developed for the high multiplicity-of-infection (MOI) scCRISPR analysis data, they were not examined in this study, which used low MOI data.

**Methods comparison using dyngen.** To benchmark the methods, simulated data were generated using dyngen, a GRN-based simulator of scRNA-seq data. A total of 750 GRNs, consisting of 100 genes, were generated by setting num_tfs = 100. In detail, 250 GRNs were generated for each of the three backbones (linear, converging, and bifurcating conversing) defined in dyngen. We used the backbones with only one steady state because they are cases similar to the real data of hiPS cells we obtained in this study.

The ground-truth GRNs were used for the simulation by dyngen. Initially, the simulation was run without KO for simtime_from_backbone(backbone) time to obtain a steady state for each backbone. Subsequently, a gene was knocked out, and the simulation was run for 100 steps from the steady state. After the KO, a total of 100 cells were sampled at four time points in regular intervals. The parameter values used in dyngen are presented in Supplementary Table 6.

We ran the simulation knocking out each of the 100 genes in each GRN and obtained expression data of 100 genes sampled from 100 cells under 100 KOs. Note that here we performed a single-gene KO multiple times. For each GRN, the expression data subset was constructed by extracting the cells corresponding to the KO genes included in the randomly selected set $M$ of genes. For each backbone, the 250 GRNs were divided into 5 sets, each of which included 50 GRNs. GRNs in each set have a different size $M$ ($|M| = 20, 40, 60, 80, 100$). The ratio of KO genes for each data set is $\frac{|M|}{100}$. We found that in some GRNs of bifurcating converging backbone, single-gene KO does not cause substantial expression variation, possibly due to the GRN structure (Supplementary Fig. 3). The amount of expression variation caused by single-gene KO (MIMOSCA score) was calculated using the $G \times G_{ko}$ matrix $\beta$ calculated by MIMOSCA as follows:

$$\text{MIMOSCA\_score} = \frac{\sum_{i,j} |\{\beta\}_{i,j}|}{G_{ko}}. \qquad (11)$$

Since RENGE assumes that single-gene KO causes a substantial amount of expression variation, we excluded GRNs with MIMOSCA_score < 2. Consequently, we used 248 GRNs for linear backbones, 233 GRNs for converging backbones, and 133 GRNs for bifurcating converging backbones, resulting in a total of 614 GRNs. To normalize the count data generated by dyngen and stabilize variance, we applied sctransform of Seurat[38]. The resulting data were used to infer GRNs by each method. The results for all the 750 GRNs are shown in Supplementary Fig. 2.

To evaluate the agreement between the inferred GRN and the ground-truth GRN, we first calculated the agreement of the presence and absence of regulation using the AUPRC ratio, while ignoring the sign of the regulation. AUPRC is a common metric that measures the agreement between the inferred and ground-truth GRNs. The AUPRC ratio is the AUPRC divided by that of a random predictor, and it was averaged for all GRNs and $M$ KO gene sets for each KO gene ratio. The AUPRC ratio for each of

the positive and negative regulations was then calculated as follows: for positive regulations the confidence level of regulation was set to 0 if it was negative, and only positive regulations were considered; negative regulation was similarly calculated.

**Methods comparison using hiPSC data.** We selected the genes to be included in the GRN of hiPSCs as follows. Let $\beta_{d2}$ be the coefficient matrix obtained by applying MIMOSCA to the day 2 cell population. $\{\beta_{d2}\}_{i,j}$ represents the expression variation of gene $i$ when gene $j$ is knocked out. The expression variation score $v_i$ of gene $i$ was defined as $v_i = \sum_j |\{\beta_{d2}\}_{i,j}|$, and the top 80 non-KO genes with large $v_i$ were selected. A total of 103 genes with 80 non-KO genes and 23 KO genes constituted the node set for the focal system in this study.

The ChIP-Atlas, a database for ChIP-seq data, was used to validate the GRN inferred from the hiPSC data. ChIP-seq data for 19 genes from human pluripotent stem cells was obtained. We used cell types included in the cell-type class "Pluripotent stem cell" defined in the ChIP-Atlas that did not contain "derived" in the cell type name. Note that the data labeled as ChIP-seq data for RUNX1T1 in ChIP-Atlas was excluded because it was actually ChIP-seq data for RUNX1-ETO. The 19 genes with ChIP-seq data consisted of 9 KO genes and 10 non-KO genes (Supplementary Table 3). The confidence level for the binding of a TF to DNA is expressed as $-10 \times \log_{10}(\text{MACS2 } q\text{-value})$. If the confidence level of the binding of gene $j$ to gene $i$ in the region of TSS $\pm$ 10kb was higher than the predetermined ChIP threshold, we assumed that regulation occurred from gene $j$ to gene $i$. This means that the ground-truth network depends on the ChIP threshold; the higher the ChIP threshold, the more reliable the regulations in the ground-truth network. We calculated the AUPRC ratio for the ground-truth GRNs of various confidence levels changing the ChIP threshold from 0 to the maximum confidence value in the data.

The rank correlation coefficient between the confidence level of each regulation was calculated using each method and the confidence level of the ChIP-seq data ($-10 \times \log_{10}(\text{MACS2 } q\text{-value})$). For RENGE, MIMOSCA, and scMAGeCK, we used $-\log_{10}(q\text{-value})$ as the confidence level, and for GENIE3, dynGENIE3, and BINGO, we used the output value of each tool itself (confidence values or weights).

We examined the details of the inferred regulations for each method by comparing it with the ground-truth network with the ChIP threshold = 300. There were 237 regulations, the same number that was observed in the ground-truth network, that were extracted for the GRNs inferred by each method, in order of confidence score of the regulation. These regulations were classified as follows. Suppose the regulation from gene $j$ to gene $i$ was inferred. If the length $k$ of the shortest path from gene $j$ to gene $i$ in the ground-truth network was 1, it was classified as direct; while if $k > 1$, it was classified as indirect. If there was no path from gene $j$ to gene $i$, it was classified as no path.

**Analysis of network.** Having inferred the GRN of 103 genes by RENGE, we focused on regulation with FDR < 0.01 and calculated the out-degree for each gene which is shown in Fig. 5b. Using this GRN, we validated our hypothesis that gene pairs with a similar set of target genes are likely to form a protein complex. Using the regulatory coefficient matrix $\mathbf{A}$ estimated by RENGE, the regulatory correlation coefficients were calculated for all gene pairs in the network as follows:

$$R = \{cor_{sp}(\mathbf{A}_{:,i}, \mathbf{A}_{:,j}) | 1 \le i, j \le G\}, \qquad (12)$$

where $\mathbf{A}_{:,i}$ denotes the $i$-th column of $\mathbf{A}$ and $cor_{sp}(\mathbf{x}, \mathbf{y})$ denotes the Spearman's rank correlation coefficient between $\mathbf{x}$ and $\mathbf{y}$. If

$cor_{sp}(\mathbf{A}_{:,i}, \mathbf{A}_{:,j})$ is close to 1, gene $i$ and gene $j$ regulate the same genes in the same direction, and if close to -1, they regulate the same genes in the opposite direction.

We compared the regulatory correlation with the protein complex data from the three databases. First, curated complexes were obtained from the CORUM3.0 database. We used all complexes in which at least 66% of their component genes were included in the 103 genes in the GRN[15]. When a gene pair was included in the same complex, the gene pair was assigned to be in the CORUM complex. Second, protein-protein interaction scores were obtained from the v11.5 of STRING (9606.protein.physical.links.v11.5.txt.gz). The protein-protein interaction scores for gene $i$ and gene $j$ are denoted as $PPI_{i,j}$. Among the gene pairs in $R$, those with $PPI_{i,j} = 0$ were assigned "STRING score low," and those with the top 10% of $PPI_{i,j}$ among gene pairs with $PPI_{i,j} > 0$ were assigned "STRING score high". Third, colocalization scores for the DNA binding of TFs were obtained from the ChIP-Atlas, using data for the cell type class of pluripotent stem cells.

**Predicting expression changes following a gene knockout**. Let $D_S = \bigcup_{t=1}^{4}(\mathbf{X}_{S,t}, \mathbf{E}_{S,t})$ be a data set containing control cells and cells in which genes in the gene set $S$ are knocked out, and $O = \{1, \cdots, 23\}$ be the indices of the genes knocked out in the hiPSC data. We trained the RENGE model using the dataset $D_{O\setminus\{j\}}$ excluding cells in which the gene $j$ ($j = 1, \cdots, 23$) was knocked out. The trained RENGE model was then used to predict the expression changes of the other genes when gene $j$ was knocked out. We calculated the Pearson correlation coefficient between the predicted and measured expression changes for the gene $j$ KO using $D_{\{j\}}$.

**Statistics and reproducibility**. All the underlying statistical details were provided earlier in the Methods section.

**Reporting summary**. Further information on research design is available in the Nature Portfolio Reporting Summary linked to this article.

## Data availability
The scCRISPR analysis data of hiPSCs were deposited into the Gene Expression Omnibus (GEO) database under the accession number GSE213069 and are available at https://www.ncbi.nlm.nih.gov/geo/query/acc.cgi?acc=GSE213069. The plasmids generated in this study were deposited with Addgene under the ID numbers found in Supplementary Data 1. The source data for the graphs in the main figures are available as Supplementary Data 2. Supplementary Tables 1 and 2 are also available as Supplementary Data 1 in an Excel file.

## Code availability
The python implementation of RENGE is available at https://github.com/masastat/RENGE. The version of the code described in this paper was deposited in Zenodo[43].

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

## Acknowledgements

This study was supported by JST CREST Grant number JPMJCR1922 (M.E., A.M. and K.Y.), JSPS KAKENHI Grant numbers JP17K20146 and JP21H04959 (K.Y.), JP20K15714 and JP22K06237 (Y.T), 17K00398 and 20K12059 (H.K.), 19H05670 and 19H03196 (A.M.), the Mitsubishi foundation (K.Y.), Takeda Science Foundation (Y.T.), Kato Memorial Bioscience Foundation (Y.T.), and Joint Usage/Research Center program of Institute for Life and Medical Sciences, Kyoto University (A.M. and H.K.). Sequencing was supported by Single-cell Genome Information Analysis Core (SignAC) at WPI-ASHBi, Kyoto University. Computations were performed using the super-computing resource provided by Human Genome Center, the Institute of Medical Science, the University of Tokyo (http://sc.hgc.jp/shirokane.html). M.I. would like to thank Naoto Yamaguchi for helpful comments regarding development of RENGE.

## Author contributions

M.I. designed and implemented RENGE and performed analyses using RENGE. H.K., A.M., Y.Y. and Y.K. contributed to developing RENGE. M.E. contributed to experimental design of CRISPR. Y.M., S.S., Y.T., and K.Y. performed CRISPR and iPSC experiments. Y.S. and N.T. performed scRNA-seq experiments. M.I. F.W. and M.E. processed and analyzed the data. M.I. and K.Y. wrote the manuscript with assistance from other authors.

## Competing interests

The authors declare no competing interests.
