## [Peer Review File · Communications Biology]

Reviewers' comments:

Reviewer #1 (Remarks to the Author):

This paper proposes RENGINE, a novel method for inferring GRNs using time series gene expression datasets in response to scCRISPR knockout, such as those generated by Perturb-seq or CROP-seq. The basis of this method is that the change in expression for each gene can be modeled as a function of the combined effect of paths of length at most some integer K' from the knockout gene in an inferred GRN. The authors evaluated RENGINE using both simulated data (generated by *dyngen*) and experimental data, and the performance was compared against representative existing methods. These methods included both those that incorporate the knowledge of gene knockouts (such as *sgMaGECK*) and those that infer based only on gene expression (*dynGENIE3*). The authors demonstrated the interpretability of the method in the biological application of hiPSCs, and select TFs identified in the inferred GRN were supported by the literature.

MAJOR COMMENTS

In the description of the algorithm, the authors use E to denote the gene expression vector in a cell at a specific time point. I take this vector to denote the measured expression values. However, in other parts of the manuscript, especially in motivating the algorithm, the authors talk about representing the "change in expression" after a knock-out. Their algorithm makes sense to me only if E denotes the vector of gene expression changes after a knock-out (since *Xg* records only the change in the expression of the knocked-out gene g). So what is the precise definition of E ?

Page 4, lines 83-84: "However, the time K' the model (1) and the measuring time t of gene expression are usually different."

Please be more clear in the text describing the relationship between K' and time. K' is first defined as the max path length in a GRN, but then equated to time in the text. Perhaps this should be rephrased as "However, the longest path length of regulatory interactions that have occurred, K' , is usually unknown at time t ".

Please use consistent notation between equations in the results section and the methods section
- M_g is not included in Equation (1) or mentioned in the results. Please explain that Equation (3) is the extension of Equation (1), or Equation (1) is a simplification of Equation (3) to introduce the method.

- The notation $E_{\{g,K'\}}$ is used in Equation (3) while the notation $E_{\{K',g\}}$ is used in equation (1)
- The $E_{\{t,c\}}$ and $M_{\{t,c\}}$ notation introduced in the methods is confusing when compared to the earlier $E_{\{K',g\}}$ and M_g because $t \neq K'$ and $c \neq g$. Please introduce new variables or otherwise mark $E'_{\{t,c\}}$ and $M'_{\{t,c\}}$ to further highlight that there is a conceptual difference between $E_{\{K',g\}}$ and $E_{\{t,c\}}$.

Page 18, lines 434-436: "The four GRNs, consisting of 100 genes, were generated by setting $\text{num tfs} = 100$ using three backbones (linear, converging, and bifurcating converging) defined in *dyngen* with only one steady state."

-Please explain why RENGINE can only infer GRNs with one steady state, or otherwise evaluate the performance against networks with multiple steady states such as bifurcating.

-Please clarify which 4 GRNs were used for the 3 backbones. Does this mean that 4 GRNs were generated for each backbone?

-To further demonstrate robustness, please evaluate against a larger number of simulated GRNs, such as 100.

Page 16, lines 414-417: "BINGO ... It can also handle KO information. In this study, the time-series data for BINGO was constructed in the same way as for *dynGENIE3*."

-Please explain why KO genes are not incorporated in the input to BINGO if it is able to handle KO information.

MINOR COMMENTS

Page 2, lines 32-34: "The most reliable way to infer a causal relationship between genes is to

examine the changes in expression resulting from perturbed gene expression"

-Please rephrase or provide a citation or further support for perturbed gene expression being "the most reliable" method.

Page 3, lines 71-72: The sentence "Here, if the effect of a change in the expression of gene j is propagated to gene i via the $k-1$ genes ..." is unclear.

-A better phrasing would be "Here, if the effect of a change in the expression of gene j is propagated to gene i via a path containing $k-1$ intermediate genes ...".

Page 12, line 335: "The GRNs of the TFs were investigated."

-What does this statement mean? Does it mean that TFs were the only nodes considered for inclusion in GRNs? If so, it would be better to make the phrasing clear.

Page 12, line 339: The elements of X_g are not clear. Does it have a non-zero value only in the dimension for the knocked-out gene g and is the value in this dimension the expression level of gene g when it is knocked out? Or could all the elements of X_g have non-zero values depending on the expression levels of the corresponding genes when g is knocked out? I wrote this comment before reading Section 4.6, which makes it clear that X_g has a non-zero value only for gene g . Making the language explicit in line 339 will be helpful to the reader.

Page 5, lines 123-124: "It is well established that the core regulatory network that maintains pluripotency is composed of POU5F1, SOX2, NANOG, and PRDM14"

-Please provide a citation explaining how this was established.

Page 5, lines 127-128: "We therefore selected 23 genes (TFs) that are thought to be involved in hiPSC pluripotency"

-Please provide a citation explaining why the selected TFs are thought to be involved.

Page 8, lines 148-149: "top 80 TFs with the largest expression changes due to gene KO"

-Please clarify in this text that "largest expression changes" specifically refers to the p-value computed by MIMOSA, instead of a different metric like absolute difference in expression, normalized variance, or other p-value. This is later explained in methods.

Page 7, lines 162-163: "These results imply a high correlation between the q-values of the regulations inferred by RENGE and those of the ChIP-seq data"

Please rephrase as "higher correlation than compared methods" or provide a threshold for "high correlation".

Page 7, lines 171-173: "However, considering that TF binding may not necessarily mean regulation, i.e., false positives can be acceptable, we also compared the methods focusing only on the proportion of the inferred regulations that were supported by TF binding, i.e., precision"

Please explain why false positives are equivalent to non-regulatory TF binding, and then acceptable. Based on my understanding, this description of an edge in the ChIP-seq network indicating binding that is not present as a regulatory edge in the inferred GRN would be a false negative rather than a false positive.

Page 8, line 192: "The top 20 genes"

-Please clarify in this text the top 20 genes are ordered by the number of out edges in the predicted GRN. This is later explained in the Figure 5b caption.

Page 14, line 370: " $m_{t,c}$ is used to ignore the squared error of KO gene $g_{t,c}$ expression in cell c at time t "

Please clarify in this text that the prediction for $g_{t,c}$ is ignored in the calculation of loss because RENGE is unable to infer self regulation.

Page 14, line 371-373: "To suppress the magnitude of each element of not only A but also A^k ($k \geq 2$), an L2 regularization term was added for A^k ($k = 1, \dots, K$)"

-Please explain why only L2 regularization is used for A^k while both L1 and L2 regularization are used for A ?

-For comparison in Figure 2b, 2d, and 2f, all scMAGECK/MIMOSCA predictions could be assumed to be regulation by non-KO genes as they are assumed to be regulation by KO genes in Figure 2a, 2c, 2e.

-To improve readability, please increase the font size in Figure 5, and either increase the line width or use a different method than solid or dashed lines to differentiate edges with and without ChIP support.

Reviewer #2 (Remarks to the Author):

The authors present a new method to infer gene regulatory network (GRN) based on time-series data from gene knockout experiments. They compare the performance of their method against that of existing approaches, on simulated data and on experimental data. The method relies on iterating the GRN by matrix exponentiation, i.e. to "walk" a series of steps in the GRN graph, and thereby be sensitive to both direct as well as indirect interactions.

The paper is clearly written, and it appears that the authors' method (RENGE) outperforms other methods on simulated and real datasets. Interestingly, the authors appear to have found a new protein complex using hiPSC data. (Though this is not independently validated). Overall, this seems to be a very useful contribution to a rapidly changing field.

Comments:

1. I found the text to be rather heavy in details. Some of these would be relevant to readers specifically interested in iPSCs, but I suspect this paper is being targeted at a broader readership.
2. It did not become evident why the performance of RENGÉ was so good. Particularly in the simulated dataset, is it because the assumptions underlying RENGÉ better match the simulator? In the discussion the authors point to other factors such as cooperative effects. Were these included in the simulation?
3. The authors correctly detect positive feedback loops that could lead to sustained states. What happens in the case of effective-negative feedback loops that lead to oscillations?

Reviewer #3 (Remarks to the Author):

Summary

In this manuscript, the authors present a computational model, RENGÉ, to infer Gene Regulatory Networks out of time-series of scRNA-seq data after various gene KO.

The highlight of this method is that it takes into consideration the fact that there are changes in expression that vary over time, which can be direct and indirect to the perturbations (single cell KOs).

The authors show the efficiency of their algorithm in different ways.

They compare the "precision" of their inference method with other methods (which are also well and fairly described). The different methods were used to infer a simulated (scRNA-seq) time-series from networks after KO and were used on actual experimental data from scCRISPR on hiPSCs to infer a pluripotency network.

They use various state-of-the-art tools to construct the benchmark networks and the ground truth network for the hiPSCs.

The comparisons using as benchmarks the simulated networks show the inference capabilities of their method, being the best one to identify regulation by KO genes, and second for non-KO genes, as a function of the ratio of KOs in the network (extensive but not overwhelming results are presented in the SI as to see comparisons in different networks).

To test RENGINE with the experimental data, as a benchmark they constructed a network using Chip-seq data of the genes in the pluripotent stem cells database. They consider the limitations of the ChIP-seq data (as it represents the binding of TF, and not necessarily actual regulation), by showing the precision of the methods as a function of the "TF significance" threshold. RENGINE showed the best performance and precision than the other methods for most of the conditions. The authors go further in showing the biological significance and observations that can be taken from their inference network (not leaving behind what can also be implied from the Chip-seq data). They observe that among the genes with the most output degree are pluripotency-related genes, which also form a positive feedback loop. And detected negative regulation from genes that are important for cell differentiation.

They also hypothesize in the formation of protein complexes given the similarities in the target genes of some gene pairs, obtained from their inferred GRN. Finally, they evaluate the power of prediction of the expression changes induced by KOs, by comparing the results of the inferred networks in which not all the gene KOs were included and when all data was used. The correlations in the expression of changes were mostly positive for the different days.

My only reservation on the analysis about the binding position being co-localized for pair of genes that form complexes and have large regulatory correlation functions. (Last part of section 2.5, Fig. 6d). They state that "Gene pairs with highly co-localized binding positions on the genome tended to have large absolute regulatory correlation and high STRING scores". I don't see a clear correlation between such variables. Nevertheless, there seem to be cases in which it is true. Then, instead of looking as a general property for the inference method, highlighting the observation and the specific pair of genes that are colocalized would be fairer.

A minor question comes from Fig. 4a and 4b.: the AUPRC ratio show abrupt changes for almost all the different methods, for specific threshold. Do the authors have an explanation for this? Are there clusters of genes that are sensitive to the same threshold?

In general, I find the paper to be clear and thorough. They demonstrate good capabilities of their method that are worth communicating and used as a tool for the identification of GNR, also a necessary one, given the good amount of RNA-seq data that is being extracted in diverse areas.

point-by-point response to reviewers

Correction

In our benchmark and GRN analyses of human iPSCs data, we used ChIP-seq data obtained from CHIP-Atlas. We found that the data labeled as ChIP-seq data of RUNX1T1 from pluripotent stem cells in this dataset actually corresponds to ChIP-seq data of RUNX1T1 and RUNX fusion protein from cells that had differentiated from pluripotent stem cells, as described in the original paper. Therefore, we excluded the RUNX1T1 ChIP-seq data (see page 18, lines 499-500, and Table S3) and reanalyzed the data, and we found our conclusions were unaffected (Figures 4, 5, and 6).

Response to reviewers

We would like to thank the reviewers for their insightful comments. Below, we show our point-by-point responses to each of the comments.

Dear Mr Ishikawa,

Your manuscript entitled "Inference of gene regulatory networks using time-series single-cell RNA-seq data with CRISPR perturbations" has now been seen by 3 referees. You will see from their comments below that while they find your work of considerable interest, some important points are raised. We are interested in the possibility of publishing your study in Communications Biology, but would like to consider your response to these concerns in the form of a revised manuscript before we make a final decision on publication.

We therefore invite you to revise and resubmit your manuscript, taking into account the points raised.

Please highlight all changes in the manuscript text file.

We are committed to providing a fair and constructive peer-review process. Do not hesitate to contact us if you wish to discuss the revision in more detail or if there are specific requests from the reviewers that you believe are technically impossible or unlikely to yield a meaningful outcome.

At the same time, we ask that you ensure your manuscript complies with our editorial policies. Please see our revision file checklist for guidance on formatting the manuscript and complying with our policies. You will also find guidelines for replying to the referees' comments. You may also wish to review our formatting guidelines for final submissions here.

When submitting the revised version of your manuscript, please pay close attention to our Digital Image Integrity Guidelines.

We would expect revisions of this nature to take around three months, but appreciate that every situation is unique. We look forward to receiving your revised manuscript when it is ready, and will

not enforce a hard deadline on this revision.

Please do not hesitate to contact me if you have any questions or would like to discuss these revisions further. We look forward to seeing the revised manuscript and thank you for the opportunity to review your work.

Best regards,

Eve

*Eve Rogers, PhD
Associate Editor, Communications Biology
4 Crinan Street
London N1 9XW, UK*

Referee expertise:

Referee #1: Benchmarking algorithms, GRNs, single-cell transcriptomics

Referee #2: Genomics

Referee #3: Computational biology

Reviewers' comments:

Reviewer #1 (Remarks to the Author):

This paper proposes RENGINE, a novel method for inferring GRNs using time series gene expression datasets in response to scCRISPR knockout, such as those generated by Perturb-seq or CROP-seq. The basis of this method is that the change in expression for each gene can be modeled as a function of the combined effect of paths of length at most some integer K from the knockout gene in an inferred GRN. The authors evaluated RENGINE using both simulated data (generated by dyngen) and experimental data, and the performance was compared against representative existing methods. These methods included both those that incorporate the knowledge of gene knockouts (such as sgMaGECK) and those that infer based only on gene expression (dynGENIE3). The authors demonstrated the interpretability of the method in the biological application of hiPSCs, and select TFs identified in the inferred GRN were supported by the literature.

Thank you for the precise understanding.

MAJOR COMMENTS

- 1. In the description of the algorithm, the authors use E to denote the gene expression vector in a cell at a specific time point. I take this vector to denote the measured expression values. However, in other parts of the manuscript, especially in motivating the algorithm, the authors talk about representing the "change in expression" after a knock-out. Their algorithm makes sense to me only if E denotes the vector of gene expression changes after a knock-out (since Xg records only*

the change in the expression of the knocked-out gene g). So what is the precise definition of E?

Thank you for the comment. E denotes the gene expression vector (not “change”). $b_{K'}$ denotes the expression vector in the wild type, and $\sum_{k'=1}^{K'} A^{k'} X_g$ represents expression change from the wild type due to KO of gene g . We have added an explanation in page 4, line 81, and page 13, lines 361-362.

2. *Page 4, lines 83-84: "However, the time K' the model (1) and the measuring time t of gene expression are usually different."
Please be more clear in the text describing the relationship between K' and time. K' is first defined as the max path length in a GRN, but then equated to time in the text. Perhaps this should be rephrased as “However, the longest path length of regulatory interactions that have occurred, K' , is usually unknown at time t ”.*

As suggested, we have rephrased the sentence to “However, the longest path length of regulatory interactions that have occurred, K' , is usually unknown at the measurement time t of gene expression.” in page 4, lines 85-86.

Please use consistent notation between equations in the results section and the methods section

3. *- M_g is not included in Equation (1) or mentioned in the results. Please explain that Equation (3) is the extension of Equation (1), or Equation (1) is a simplification of Equation (3) to introduce the method.*

We have added an explanation that Equation (3) is the extension of Equation (1) in page 13, lines 359-360.

4. *- The notation $E_{\{g,K'\}}$ is used in Equation (3) while the notation $E_{\{K',g\}}$ is used in equation (1)*

We have changed the notation to $E_{\{g,K'\}}$ in page 13, Equation (3).

5. *- The $E_{\{t,c\}}$ and $M_{\{t,c\}}$ notation introduced in the methods is confusing when compared to the earlier $E_{\{K',g\}}$ and M_g because $t \neq K'$ and $c \neq g$. Please introduce new variables or otherwise mark $E'_{\{t,c\}}$ and $M'_{\{t,c\}}$ to further highlight that there is a conceptual difference between $E_{\{K',g\}}$ and $E_{\{t,c\}}$.*

We have marked $E'_{\{t,c\}}$ and $M'_{\{t,c\}}$ in page 13 and Equation (5). We have also added an explanation for $M'_{\{t,c\}}$ in page 14, line 381.

Page 18, lines 434-436: "The four GRNs, consisting of 100 genes, were generated by setting $num\ fs = 100$ using three backbones (linear, converging, and bifurcating converging) defined in dynngen with only one steady state."

6. *-Please explain why RENGINE can only infer GRNs with one steady state, or otherwise evaluate the performance against networks with multiple steady states such as bifurcating.*

Thank you for the useful comment. RENGINE is fundamentally applicable to a network with multiple steady states. In such cases, the sample at a given time may contain cells derived from different steady states. Cells at a time can be clustered to each trajectory starting from each steady state. RENGINE can be applied to each trajectory by allowing β_t to take different values for each trajectory. However, it is beyond the scope of this research because the real data of human iPS cells we obtained in this research is the case of a single steady state. To make it clearer, we have added an explanation in page 17, lines 459-460.

7. *-Please clarify which 4 GRNs were used for the 3 backbones. Does this mean that 4 GRNs were generated for each backbone?*

Since one GRN was generated for each backbone, "a total of three GRNs" is correct. "four GRNs" is a typo. This point has been addressed in the response to comment 8 below.

8. *-To further demonstrate robustness, please evaluate against a larger number of simulated GRNs, such as 100.*

Increasing the number of GRNs per backbone to 100 is impractical because it would take too much time in the computing environment available to us. This is because the current approach requires, for each GRN, simulating the dynamics for a total of 50 sets of KO genes and performing GRN inference with each of the seven methods.

Therefore, to increase the robustness of the results, we changed the benchmarking approach. We have generated 250 GRNs per backbone and assigned one set of KO genes per GRN (not 50 sets per GRN), resulting in a total of 750 GRNs. We found that in some GRNs of bifurcating converging backbone, single-gene KO does not cause substantial expression variation, possibly due to the GRN structure. Since RENGINE assumes that single-gene KO causes a substantial amount of expression variation, we excluded GRNs with low expression variation. Consequently, we used 248 GRNs for linear backbones, 233 GRNs for converging backbones, and 133 GRNs for bifurcating

converging backbones, resulting in a total of 614 GRNs. The detailed method is described in page 17 and 18.

The obtained results were consistent with the original results for regulation by non-KO genes (Figure 2b,d,f). For regulation by KO genes, RENGE showed performance comparable to, but not significantly superior to, existing methods (Figure 2a,c,e). Therefore, the claim in the main text has been changed accordingly in page 5. Also, we have added a sentence, "Though RENGE showed superior performance on average, the performance of RENGE depended on the backbone used, which may be due to the complexity of the GRN." in page 5, lines 119-121, to explain the difference in performance by backbone (Figure S1). The results for all 750 GRNs without filtering were also shown in Figure S2.

9. Page 16, lines 414-417: "BINGO ... It can also handle KO information. In this study, the time-series data for BINGO was constructed in the same way as for dynGENIE3."
-Please explain why KO genes are not incorporated in the input to BINGO if it is able to handle KO information.

KO genes were incorporated in the input to BINGO. We have added an explanation to page 16, lines 438-440, as "BINGO takes two types of input data, time-series expression data (as data.ts) and KO gene data (as data.ko). The time-series data was constructed in the same way as for dynGENIE3, and KO gene data was constructed based on gRNA assignment."

MINOR COMMENTS

10. Page 2, lines 32-34: "The most reliable way to infer a causal relationship between genes is to examine the changes in expression resulting from perturbed gene expression"
-Please rephrase or provide a citation or further support for perturbed gene expression being "the most reliable" method.

We have added a citation and rephrased as "The most reliable way to infer a causal relationship between genes would be to examine the changes in expression resulting from perturbed gene expression" in page 2, line 33.

11. Page 3, lines 71-72: The sentence "Here, if the effect of a change in the expression of gene j is propagated to gene i via the $k-1$ genes ..." is unclear.

-A better phrasing would be “Here, if the effect of a change in the expression of gene j is propagated to gene i via a path containing $k-1$ intermediate genes ...”.

We have changed the phrase as suggested by the reviewer’s comment in page 3, line 73.

12. Page 12, line 335: *“The GRNs of the TFs were investigated.”*

-What does this statement mean? Does it mean that TFs were the only nodes considered for inclusion in GRNs? If so, it would be better to make the phrasing clear.

We have changed the phrase to “We investigated GRNs whose nodes were TFs only.” in page 13, line 348.

13. Page 12, line 339: *The elements of \mathbf{X}_g are not clear. Does it have a non-zero value only in the dimension for the knocked-out gene g and is the value in this dimension the expression level of gene g when it is knocked out? Or could all the elements of \mathbf{X}_g have non-zero values depending on the expression levels of the corresponding genes when g is knocked out? I wrote this comment before reading Section 4.6, which makes it clear that \mathbf{X}_g has a non-zero value only for gene g . Making the language explicit in line 339 will be helpful to the reader.*

We have changed the phrase to “ \mathbf{X}_g is a G -dimensional vector of which g th component is the expression change of gene g due to its KO, and the other components are zero.” in page 13, lines 353-354. To make it more clear, we have added an explanation of the variables in page 13, lines 348-349, 354-355.

14. Page 5, lines 123-124: *“It is well established that the core regulatory network that maintains pluripotency is composed of POU5F1, SOX2, NANOG, and PRDM14”*
-Please provide a citation explaining how this was established.

We have added citations in page 5, line 130.

15. Page 5, lines 127-128: *“We therefore selected 23 genes (TFs) that are thought to be involved in hiPSC pluripotency”*
-Please provide a citation explaining why the selected TFs are thought to be involved.

We have added citations in page 6, line 134.

16. Page 8, lines 148-149: *“top 80 TFs with the largest expression changes due to gene KO”*
-Please clarify in this text that “largest expression changes” specifically refers to the p-value computed by MIMOSA, instead of a different metric like absolute difference in expression, normalized variance, or other p-value. This is later explained in methods.

“largest expression changes” specifically refers to the coefficient β computed by MIMOSCA, not the p-value. The coefficient β is similar to (but not the same as) the mean absolute difference in expression. We have added a sentence “The expression changes were calculated using the coefficient matrix β obtained by applying MIMOSCA” in page 6, lines 154-155.

17. Page 7, lines 162-163: *"These results imply a high correlation between the q-values of the regulations inferred by RENGE and those of the ChIP-seq data"*
Please rephrase as “higher correlation than compared methods” or provide a threshold for “high correlation”.

We have rephrased as “higher correlation than compared methods” in page 7, line 168.

18. Page 7, lines 171-173: *"However, considering that TF binding may not necessarily mean regulation, i.e., false positives can be acceptable, we also compared the methods focusing only on the proportion of the inferred regulations that were supported by TF binding, i.e., precision"*
Please explain why false positives are equivalent to non-regulatory TF binding, and then acceptable. Based on my understanding, this description of an edge in the ChIP-seq network indicating binding that is not present as a regulatory edge in the inferred GRN would be a false negative rather than a false positive.

Thank you for the careful comment. “false negatives” is correct. We have corrected the phrase to “false negatives in a GRN inferred by RENGE can be acceptable” in page 7, 178.

19. Page 8, line 192: *"The top 20 genes"*
-Please clarify in this text the top 20 genes are ordered by the number of out edges in the predicted GRN. This is later explained in the Figure 5b caption.

We have rephrased as “The top 20 genes out of 103 genes (19%) ordered by the number of out-edges are associated with 51% of the regulations detected, ...” in page 8, line 199.

20. Page 14, line 370: *" $m_{t,c}$ is used to ignore the squared error of KO gene $g_{t,c}$ expression in cell c at time t "*
Please clarify in this text that the prediction for $g_{t,c}$ is ignored in the calculation of loss because RENGE is unable to infer self regulation.

The reason why the prediction for $g_{t,c}$ is ignored is not that RENGÉ cannot infer self-regulation, but rather because the observed expression level of a KO gene $g_{t,c}$ may not reflect the abundance of the functional protein; when using the CRISPR system, even when a gene is totally knocked out, the mRNA for that gene may still be expressed.

We have rephrased as “ $\mathbf{m}_{t,c}$ is used to ignore the squared error of KO gene $g_{t,c}$ expression in cell c at time t because mRNA of KO gene $g_{t,c}$ may still be expressed even when the functional protein is lost when using the CRISPR system.” in page 14, lines 388-390.

21. Page 14, line 371-373: "To suppress the magnitude of each element of not only A but also A^k ($k \geq 2$), an L2 regularization term was added for A^k ($k = 1, \dots, K$)"
-Please explain why only L2 regularization is used for A^k while both L1 and L2 regularization are used for A ?

We have added an explanation “Note that the L1 regularization term was only added for \mathbf{A} and not for \mathbf{A}^k ($k \geq 2$) because \mathbf{A} represents a GRN and thus is expected to be sparse, but \mathbf{A}^k ($k \geq 2$) is not necessarily sparse.” in page 15, lines 393-395.

22. -For comparison in Figure 2b, 2d, and 2f, all scMAGECK/MIMOSCA predictions could be assumed to be regulation by non-KO genes as they are assumed to be regulation by KO genes in Figure 2a, 2c, 2e.

scMAGECK/MIMOSCA cannot predict regulation by non-KO genes. In other words, regulations by non-KO genes are not included in the output of these methods. If we take it as that there is no regulation by non-KO genes, then its AUPRC is the same as a random predictor. Therefore, scMAGECK/MIMOSCA are not shown in Figures 2b, 2d, and 2f.

23. -To improve readability, please increase the font size in Figure 5, and either increase the line width or use a different method than solid or dashed lines to differentiate edges with and without ChIP support.

We have increased the font size in Figure 5, but it was difficult to increase the font size significantly due to space limitations. In Figure 5de, we have increased the line width and changed from dashed line to dotted line to make the difference clearer.

Reviewer #2 (Remarks to the Author):

The authors present a new method to infer gene regulatory network (GRN) based on time-series data from gene knockout experiments. They compare the performance of their method against that of existing approaches, on simulated data and on experimental data. The method relies on iterating the GRN by matrix exponentiation, i.e. to "walk" a series of steps in the GRN graph, and thereby be sensitive to both direct as well as indirect interactions.

The paper is clearly written, and it appears that the authors' method (RENGE) outperforms other methods on simulated and real datasets. Interestingly, the authors appear to have found a new protein complex using hiPSC data. (Though this is not independently validated). Overall, this seems to be a very useful contribution to a rapidly changing field.

We express our thanks to the reviewer for giving positive comments and appreciation of our work.

Comments:

1. I found the text to be rather heavy in details. Some of these would be relevant to readers specifically interested in iPSCs, but I suspect this paper is being targeted at a broader readership.

Thank you for the suggestion. We agree that this paper is being targeted at a broad readership, but we have considered the details described in the paper to be necessary because we obtained and analyzed the new real data of human iPSCs on our own in this paper. I think the description on real data analysis of our paper is not so heavy compared to other research papers of GRN inference methods.

2. It did not become evident why the performance of RENGE was so good. Particularly in the simulated dataset, is it because the assumptions underlying RENGE better match the simulator? In the discussion the authors point to other factors such as cooperative effects. Were these included in the simulation?

The benchmark results (Fig. 2,4) show that the inference methods using nonlinear models, which include cooperative effects as in the simulator are not producing better results. Rather, RENGE, which uses a simple linear model, produces higher inference accuracy on average.

The important feature of RENGE is that it can fully exploit both KO gene information and temporal information, as shown in Fig. 1d, by using a model that represents the process

of propagation of perturbation effects on the network. BINGO can also handle both KO gene information and temporal information, but it may not fully utilize the information on the KO genes, as shown in Fig. 4f and described in page 7. This may be because BINGO was developed for general time-series expression data, not necessarily with gene KO.

3. The authors correctly detect positive feedback loops that could lead to sustained states. What happens in the case of effective-negative feedback loops that lead to oscillations?

RENGE is a method to infer each of the edges in a network with a sign. In principle, it is possible to estimate each edge regardless of whether the network contains positive or negative feedback. Actually, simulation benchmarks show that networks with negative feedbacks, such as those generated using converging backbone, could be inferred with higher inference accuracy than the other methods. If the network contains negative feedback, gene expression can oscillate in wild-type cells. In this case, it is necessary to estimate the time oscillating b_t , but this is a problem that the current RENGE model can handle.

Reviewer #3 (Remarks to the Author):

Summary

In this manuscript, the authors present a computational model, RENGE, to infer Gene Regulatory Networks out of time-series of scRNA-seq data after various gene KO.

The highlight of this method is that it takes into consideration the fact that there are changes in expression that vary over time, which can be direct and indirect to the perturbations (single cell KOs).

The authors show the efficiency of their algorithm in different ways.

They compare the “precision” of their inference method with other methods (which are also well and fairly described). The different methods were used to infer a simulated (scRNA-seq) time-series from networks after KO and were used on actual experimental data from scCRISPR on hiPSCs to infer a pluripotency network.

They use various state-of-the-art tools to construct the benchmark networks and the ground truth network for the hiPSCs.

The comparisons using as benchmarks the simulated networks show the inference capabilities of their method, being the best one to identify regulation by KO genes, and second for non-KO genes, as a function of the ratio of KOs in the network (extensive but not overwhelming results are presented in the SI as to see comparisons in different networks).

To test RENGE with the experimental data, as a benchmark they constructed a network using Chip-seq data of the genes in the pluripotent stem cells database. They consider the limitations of the ChIP-seq data (as it represents the binding of TF, and not necessarily actual regulation), by showing the precision of the methods as a function of the “TF significance” threshold. RENGE showed the best performance and precision than the other methods for most of the conditions.

The authors go further in showing the biological significance and observations that can be taken from their inference network (not leaving behind what can also be implied from the Chip-seq data). They

observe that among the genes with the most output degree are pluripotency-related genes, which also form a positive feedback loop. And detected negative regulation from genes that are important for cell differentiation.

They also hypothesize in the formation of protein complexes given the similarities in the target genes of some gene pairs, obtained from their inferred GRN. Finally, they evaluate the power of prediction of the expression changes induced by KOs, by comparing the results of the inferred networks in which not all the gene KOs were included and when all data was used. The correlations in the expression of changes were mostly positive for the different days.

We thank the reviewer for the positive comments and appreciation of our work.

- 1. My only reservation on the analysis about the binding position being co-localized for pair of genes that form complexes and have large regulatory correlation functions. (Last part of section 2.5, Fig. 6d). They state that “Gene pairs with highly co-localized binding positions on the genome tended to have large absolute regulatory correlation and high STRING scores”. I don't see a clear correlation between such variables. Nevertheless, there seem to be cases in which it is true. Then, instead of looking as a general property for the inference method, highlighting the observation and the specific pair of genes that are colocalized would be fairer.*

We found there is a weak but statistically significant correlation, so we have rephrased it as “There was a weak but statistically significant correlation (correlation = 0.28, p-value < 0.0005) between the absolute regulatory correlation of a gene pair and the co-localization score of the genomic binding position of the pair.” in page 9, lines 247-249. Also, we have highlighted the specific pairs with high co-localization scores in page 9, lines 249-254 and Figure 6.

- 2. A minor question comes from Fig. 4a and 4b.: the AUPRC ratio show abrupt changes for almost all the different methods, for specific threshold. Do the authors have an explanation for this? Are there clusters of genes that are sensitive to the same threshold?*

In general, as ChIP threshold increases the ground-truth network becomes sparse. The specific values along horizontal axis indicate the points where the number of regulations in the ground-truth network decrease. The larger the ChIP threshold, the longer the interval between the points. In other words, it becomes more likely that the ground-truth regulations does not change even when the ChIP threshold changes. The change in the ground-truth network will of course influence AUPRC of all methods.

In general, I find the paper to be clear and thorough. They demonstrate good capabilities of their method that are worth communicating and used as a tool for the identification of GNR, also a necessary one, given the good amount of RNA-seq data that is being extracted in diverse areas.

REVIEWERS' COMMENTS:

Reviewer #2 (Remarks to the Author):

In this revision, the authors have included a correction (removal of certain wrongly-labeled data used in the earlier analysis). Their main conclusions remain unchanged.

Apart from this, the authors have addressed all reviewer comments, and improved the flow of the manuscript. This is a very interesting method which will stimulate interest in the GRN field.

Reviewer #3 (Remarks to the Author):

I thank the authors for their answers and clarifications to my comments.

The authors replied to all the comments made by me and the other referees and revised the manuscript accordingly.

They demonstrated that their method, RENGÉ, is novel and has a good performance to infer GRNs.

The description of the method and its capabilities are better described after their revision.

To my mind, the work seems to me to be significant and a helpful tool for the analysis and comprehension of the structure and dynamics of GRNs and single-cell RNA-seq data.

Therefore, do support to consider its publication in Communications Biology.

Reviewer #4 (Remarks to the Author):

The authors propose RENGÉ, a computational model to infer GRNs using scCRISPR data. The authors then validate RENGÉ against previously proposed computational models using both simulated single-cell data (via dyngen method), as well as using newly generated scCRISPR data on hiPSC. In brief, the authors proposed a sound computational model that was shown to outperform previously proposed methods in the literature in both simulated and real data yielding novel insight into hiPSCs,

While the authors have addressed most almost all comments from reviewers, it still presents some shortcomings:

The writing of section 1 should be improved. Previous reviewer (Reviewer #1) already noted this major shortcoming. While the authors have addressed some of Reviewer #1 comments, the notation is still confusing. For example, why use capital letters to denote both vectors and matrices (E_g and $A^{\{k\}}$)? Additionally, $E_{\{g,K\}}$ and E_g are still confusing, rather than explaining them in the text, please rephrase the notation. Also, the method section uses slightly different notation, and while explained in the text, I still find it confusing.

My other main concern is the scalability of the method. Reviewer #1 suggested to increase the size of the simulated benchmarking, and the authors argued that increasing to 100 GRNs/KO was computationally infeasible. While the authors did increase the benchmarking, I would suggest to perform a time and memory assessment of the different methods for varying number of GRNs and KO.

point-by-point response to reviewers

Response to reviewers

We would like to thank the reviewers for their insightful comments. Below, we show our point-by-point responses to each of the comments.

Reviewers' comments:

Reviewer #4 (Remarks to the Author):

The authors propose RENGINE, a computational model to infer GRNs using scCRISPR data. The authors then validate RENGINE against previously proposed computational models using both simulated single-cell data (via dyngen method), as well as using newly generated scCRISPR data on hiPSC. In brief, the authors proposed a sound computational model that was shown to outperform previously proposed methods in the literature in both simulated and real data yielding novel insight into hiPSCs,

We express our thanks to the reviewer for giving positive comments and appreciation of our work.

While the authors have addressed most almost all comments from reviewers, it still presents some shortcomings:

The writing of section 1 should be improved. Previous reviewer (Reviewer #1) already noted this major shortcoming. While the authors have addressed some of Reviewer #1 comments, the notation is still confusing. For example, why use capital letters to denote both vectors and matrices (E_g and $A^{\{k\}}$)? Additionally, $E_{\{g,K\}}$ and E_g are still confusing, rather than explaining them in the text, please rephrase the notation. Also, the method section uses slightly different notation, and while explained in the text, I still find it confusing.

Thank you for your constructive feedback. In the implementation of RENGINE, the vectors $E_{\{c,t\}}$ and $X_{\{c,t\}}$ are derived as column vectors of matrices such as the expression matrix. Using capital letters for these vectors aligns with the code, which aids in maintaining consistency between the algorithm's description and its implementation. Additionally, the lowercase 'e' is conventionally used for specific vectors, such as the standard basis vectors in mathematical contexts. Therefore, we prefer to keep E and X in capital letters.

To enhance clarity, we have revised all instances of E_g to $E_{\{g,t\}}$ in the Results section. Where the subscript includes the order of regulation K' , we now denote this with

a dash as $E'_{\{g,K\}}$ to distinguish it from instances where the subscript includes the sampling time t , which is represented as $E_{\{g,t\}}$ without a dash. Additionally, we have explicitly stated in the Methods section that the subscripts for variables such as E and X have been intentionally altered from those in the Results section (lines 378-380). We have also corrected several typographical errors in the Methods section to ensure a consistent notation throughout the manuscript.

My other main concern is the scalability of the method. Reviewer #1 suggested to increase the size of the simulated benchmarking, and the authors argued that increasing to 100 GRNs/KO was computationally infeasible. While the authors did increase the benchmarking, I would suggest to perform a time and memory assessment of the different methods for varying number of GRNs and KO.

Thank you for the suggestion. We claimed that performing inference for 100 GRNs/backbone in the benchmark is computationally infeasible, rather than 100 GRNs/KO. This stems from the fact that in the initially submitted version for benchmarking, we were performing inference for 50 sets of KO genes for each GRN. Considering there are three backbones, scaling this up to 100 GRNs per backbone would require conducting $3 \times 100 \times 50 = 15,000$ inference tasks. Despite our efforts to assess time and memory requirements, we found that because RENGE utilizes gradient descent for parameter estimation, these metrics are highly dependent on the data. Due to time constraints, we were unable to expand the dataset sufficiently to obtain stable results. Instead, we have noted this as a limitation of this study in lines 310-313.